# FAM111A regulates replication origin activation and cell fitness

Diana O Rios-Szwed[1] , Vanesa Alvarez[1] , Luis Sanchez-Pulido[2] , Elisa Garcia-Wilson[1], Hao Jiang[3] , Susanne Bandau[1], Angus Lamond[3], Constance Alabert[1]

FAM111A is a replisome-associated protein and dominant mutations within its trypsin-like peptidase domain are linked to severe human developmental syndrome, the Kenny–Caffey syndrome. However, FAM111A functions remain unclear. Here, we show that FAM111A facilitates efficient activation of DNA replication origins. Upon hydroxyurea treatment, FAM111A-depleted cells exhibit reduced single-stranded DNA formation and a better survival rate. Unrestrained expression of FAM111A WT and patient mutants causes accumulation of DNA damage and cell death, only when the peptidase domain remains intact. Unrestrained expression of FAM111A WT also causes increased single-stranded DNA formation that relies on S phase entry, FAM111A peptidase activity but not its binding to proliferating cell nuclear antigen. Altogether, these data unveil how FAM111A promotes DNA replication under normal conditions and becomes harmful in a disease context.

## Introduction

In humans, heterozygous point mutations in FAM111A are linked to two severe developmental syndromes: the Kenny–Caffey syndrome (KCS2, OMIM-127000) and Gracile bone dysplasia (GCLEB, OMIM-602361). In both diseases, patients are characterized by, among others, short stature, hypocalcemia, hypoparathyroidism, and dense or gracile bones (Welter & Machida, 2022). Heterozygous de novo mutations are the most common, and the AlphaFold–predicted structure of FAM111A reveals that patient mutations are located within two clusters: within the enzyme domain and in a flexible region between the single-stranded DNA (ssDNA)-binding domain and the enzyme domain. Remarkably, the R569H point mutation is located outside of the enzyme domain, in the C-terminal region of the FAM111A gene and is found in seven unrelated

KCS2 patients, supporting a causal effect of FAM111A mutation in KCS2 (Unger et al, 2013; Isojima et al, 2014; Abraham et al, 2017). FAM111A catalytic activity has been shown in vitro, and recent work revealed that in cellulo FAM111A exhibits autocleavage activity when its peptidase domain is intact (Hoffmann et al, 2020; Kojima et al, 2020). Interestingly, the R569H mutation, and those of three other KCS2 and GCLEB patients, Y511H, S342Del, and D528G, do not compromise but rather enhance FAM111A autocleavage activity (Kojima et al, 2020). As FAM111A functions and substrates remain unknown, it is unclear how gain-of-function mutations contribute to KCS2 and GCLEB etiology (Welter & Machida, 2022). To provide better diagnosis and management of these conditions, it is therefore fundamental to understand the role of FAM111A in normal and disease contexts.

Pioneering work suggests that FAM111A functions as a viral host range restriction factor (Fine et al, 2012) as upon SV40 viral infection, FAM111A is recruited to sites of viral replication and reduces viral replication rates (Fine et al, 2012; Tarnita et al, 2019). Similarly, FAM111A is recruited to cellular DNA replication sites and its transient overexpression blocks DNA replication (Alabert et al, 2014; Tarnita et al, 2019; Hoffmann et al, 2020). However, in absence of FAM111A, the rate of DNA synthesis is also reduced, suggesting that FAM111A may also play a positive role in DNA replication (Alabert et al, 2014). Consistent with this, FAM111A has recently been shown to promote fork progression through chemically induced DNA-binding protein crosslinks (Kojima et al, 2020). Mechanistically, FAM111A binds directly to proliferating cell nuclear antigen (PCNA) through an N-terminal PCNA-interacting protein box (PIP) (Alabert et al, 2014), and to ssDNA through an ssDNA-binding domain (Kojima et al, 2020). Thus, clues have emerged for possible new roles for FAM111A under stress conditions, yet the molecular function of FAM111A under normal conditions remains unclear. Moreover, the repertoire of FAM111A substrates has yet to be identified.

Here, we have investigated the molecular mechanisms that link FAM111A to DNA replication. We report that FAM111A supports efficient origin and dormant origin activation. Moreover, upon

---

[1]MCDB, School of Life Sciences, University of Dundee, Dundee, UK [2]MRC Human Genetics Unit, MRC Institute of Genetics and Molecular Medicine at the University of Edinburgh, Edinburgh, UK [3]MCDB, Quantitative Proteomics Laboratory, School of Life Sciences, University of Dundee, Dundee, UK

Correspondence: c.alabert@dundee.ac.uk
Diana O Rios-Szwed's present address is MRC Institute of Genetics and Molecular Medicine at the University of Edinburgh, Edinburgh, UK
Luis Sanchez-Pulido's present address is European Molecular Biology Laboratory, European Bioinformatics Institute (EMBL-EBI), Wellcome Genome Campus, Cambridgeshire, UK

hydroxyurea (HU) treatments, FAM111A-depleted cells exhibit a reduced ssDNA formation and a better survival rate. FAM111A's main interactor is its paralog FAM111B. Both paralogs are recruited to newly replicated chromatin, yet they do not appear to target each other to degradation. Furthermore, although they are epistatic in positively promoting DNA replication, FAM111A also possesses FAM111B independent functions. Overexpression of FAM111A or expression of FAM111A harboring KCS2 and GCLB2 patient mutations cause increased level of DNA damage and cell death. Notably, FAM111A overexpression leads to extensive ssDNA formation. Although increased DNA damage is a consequence of apoptosis (Hoffmann et al, 2020), ssDNA formation upon FAM111A over-expression is not caused by apoptosis, supporting that ssDNA accumulation could be one of the primary cellular stresses caused by FAM111A overexpression. Importantly, FAM111A-induced ssDNA formation requires an intact FAM111A peptidase domain and can be prevented by blocking S phase entry.

## Results

### FAM111A depletion reduces activation of licensed origins

We first examined the ability of cells to replicate in absence of FAM111A. In FAM111A-depleted cells, DNA synthesis rate is reduced, and cell proliferation is impaired (Figs 1A and B and S1A and B). Moreover, cells accumulate at the G1/S transition (Figs 1C and S1C), although the replicative and DNA repair checkpoints are not activated (Figs 1D and S1D). To determine whether the reduced DNA synthesis rate observed upon FAM111A depletion resulted from a replisome progression defect (slower forks) or a replication initiation defect (fewer forks), we analyzed DNA replication at the single-molecule level using DNA molecular combing (Conti et al, 2001). To this end, newly replicated DNA was successively pulse labeled using two nucleotide analogs CldU and IdU, and CldU signals were used to determine replisome elongation rates (Fig 1E). Replisome progression was not impaired upon FAM111A depletion, with fork speed being slightly increased instead (Figs 1E and S1E and F). In contrast, the inter-fork distance was increased in FAM111A-depleted cells, although not significantly under all conditions (Fig S1G). Therefore, to further test the possibility that fewer origins had initiated upon FAM111A depletion, we artificially triggered dormant origin activation with the CHK1 inhibitor 7-hydroxystaurosporine (UCN-01) and measured the resulting inter-fork distance (Ge et al, 2007; Maya-Mendoza et al, 2007; Petermann et al, 2010; Feng et al, 2016; Saldivar et al, 2017). As expected, the inter-fork distance was reduced in control cells upon UCN-01 treatment due to the activation of dormant origins (Fig S1G). In FAM111A-depleted cells, however, the inter-fork distance remained higher than in control cells (Figs 1F and S1H). Moreover, in absence of FAM111A, the induction of origin firing upon UCN-01 treatment was also less efficient compared with control (Fig S1I), suggesting that dormant origin firing is also impaired. Altogether, these data revealed that FAM111A is dispensable for fork progression but supports DNA replication initiation of active and dormant origins.

DNA replication initiation is a two-step process. In G1 phase, origins are licensed by the loading of MCM2-7 complexes, whereas in S phase, a fraction of the origins are activated by the CDK- and DDK-dependent recruitment of CDC45, the GINS complex, and the rest of replisome (Riera et al, 2017; Ganier et al, 2019; Marchal et al, 2019). To identify at which stage of replication initiation FAM111A may function, we first examined the origin licensing efficiency in FAM111A-depleted cells by quantifying MCM2 abundance on chromatin in G1 phase cells by quantitative image-based cytometry (QIBC) (Fig 1C and G). QIBC provides measures of the intensity of a protein by immunofluorescence, at the single-cell level and in thousands of cells, bridging the gap between microscopy and flow cytometry (Toledo et al, 2013). In FAM111A-depleted cells, MCM2 loading was not impaired (Fig 1G and H), indicating that FAM111A does not promote origin licensing. MCM2 level were instead slightly increased upon FAM111A depletion. In contrast, CDC45 abundance on S phase chromatin was reduced upon FAM111A depletion (Fig 1I and J), suggesting that FAM111A may promote origin firing. Consistent with the ability of FAM111A to facilitate dormant origin activation (Fig 1F), CDC45 recruitment to chromatin was also impaired in UCN-01–treated FAM111A-deficient cells (Fig S1J). Mirroring CDC45, chromatin-bound RPA levels were also reduced in S phase upon FAM111A depletion (Fig S1K and L) whereas the pool of available nuclear RPA was not reduced (Fig S1M). RPA level in nucleus and cell extracts were instead higher upon FAM111A depletion (Fig S1D and M). Importantly, FAM111A depletion did not activate the ATR-CHK1 pathway (Fig 1D), excluding that in FAM111A-depleted cells, origin activation was impaired indirectly through activation of the replication checkpoint pathway (Saldivar et al, 2017). Altogether, these results indicate that FAM111A depletion impairs activation of licensed origins.

### FAM111A promotes ssDNA formation upon fork stalling

Under condition of replicative stress, dormant origin activation is essential to complete genome duplication (Fig 2A) (Marchal et al, 2019). We thus hypothesized that under conditions of replicative stress, FAM111A depletion will be detrimental to cell survival. To test this prediction, we quantified the effect of FAM111A depletion on cell survival upon HU treatment which blocks the deoxynucleotide production, arrests replisomes, and provokes dormant origin firing. siRNA-transfected cells were exposed to a 24-h treatment with HU and left to recover for 14 d. Surprisingly, FAM111A-depleted cells were resistant to the HU treatment compared with control cells (Fig 2B). Consistent with this, after short term HU treatment (2 h), lower levels of DNA damage were observed in FAM111A-depleted cells (Fig 2C). Therefore, although FAM111A promotes dormant origin activation, upon short or prolonged HU treatments, FAM111A depletion protects cells against HU-induced replicative stress.

To understand this paradox, we further examined the cellular response to HU in FAM111A-depleted cells, by measuring RPA accumulation by QIBC. As expected in control cells, accumulation of chromatin-bound RPA was detectable 2 h after HU treatment (Fig S2A). Compared with control cells, FAM111A-deficient cells showed significant lower accumulation of RPA on chromatin. Similar results were observed in HU-treated cells stably expressing GFP-RPA1 (Fig 2D), cells transfected with distinct set of siRNAs (Figs 2E and S2B),

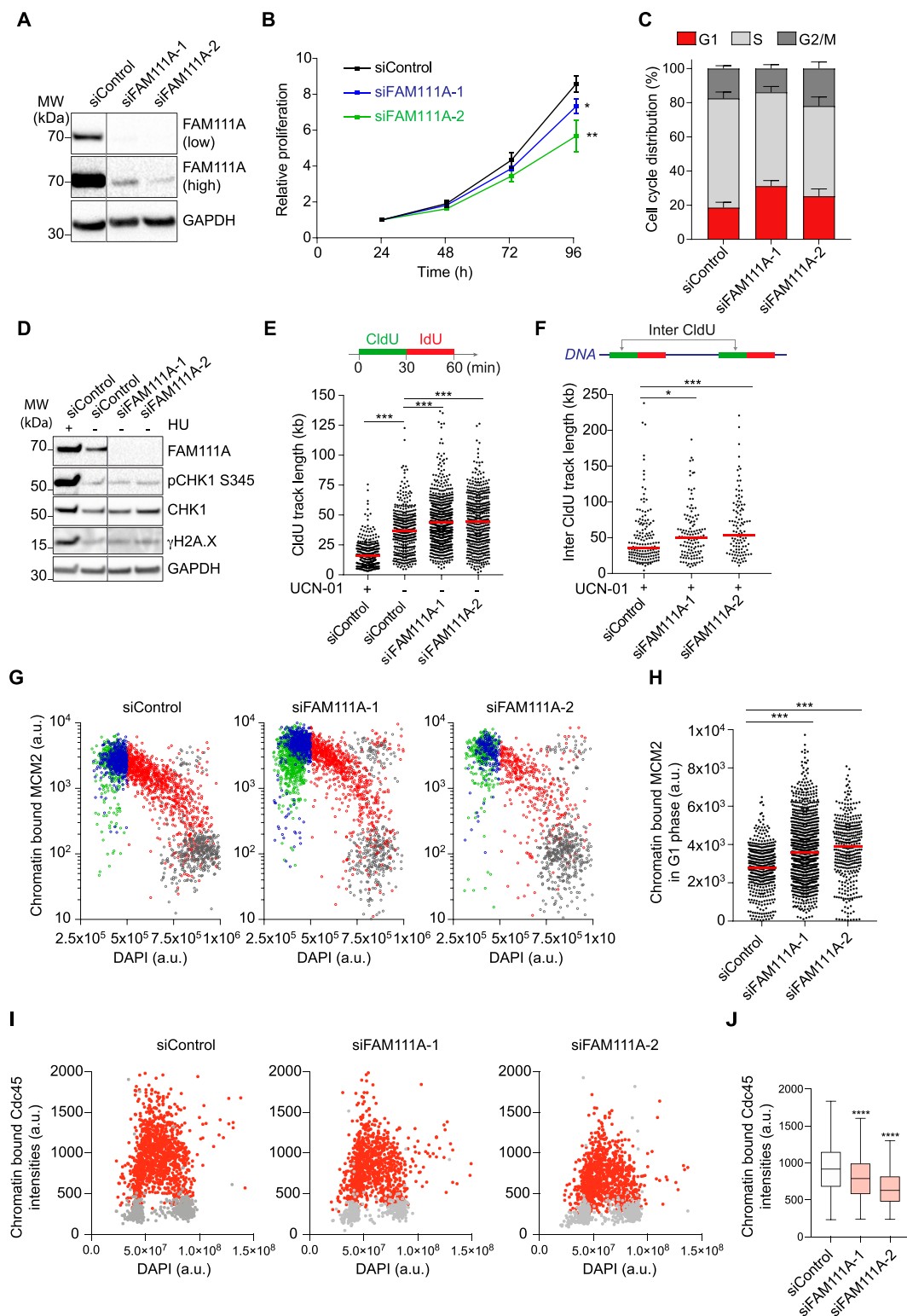

**Figure 1. FAM111A depletion impairs origin activation at the G1/S transition.**
**(A)** Immunoblot of whole cell extracts from siRNA-transfected U-2-OS cells for 48 h. **(B)** Cell proliferation of U-2-OS siRNA-transfected cells measured by Cell Titer Glo 2.0 assay. Data are represented as mean and SD of three technical replicates, n = 2. **(C)** Cell cycle distribution of U-2-OS siRNA-transfected cells detected by Quantitative Image-Based Cytometry (QIBC). EdU and DAPI contents are used to gate distinct cell cycle phases (Fig S1C). Data are represented as mean and SD of three independent experiments. **(D)** Immunoblot of whole cell extracts from siRNA-transfected U-2-OS cells for 48 h. Control cells were treated with 3 mM HU for 2 h. **(E)** Analysis of replication fork speed by DNA combing. Top, labeling strategy. Bottom, size distribution of CldU track length. Red bar represents the median; n > 421 tracks were analyzed. **(F)** Top, inter-fork distance measurement schematic. Bottom, distribution of inter CldU track length. Cells were treated with 300 nM UCN-01 for 1 h before and during

and upon treatment with the DNA polymerase-α inhibitor aphidicolin (APH) (Fig 2F). Notably, reduced RPA accumulation did not prevent the DNA replication checkpoint activation in HU-arrested cells (Figs 2G and S2C). This result could in part explain why FAM111A-depleted cells are HU resistant, as extensive ssDNA impairs cell survival (Toledo et al, 2013). As RPA accumulation is an indirect measure of increased ssDNA formation, we next directly measured ssDNA exposure level in FAM111A-depleted cells. To do so, we used BrdU labeling and detection under non-denaturing conditions (Mejlvang et al, 2014) (Figs 2H and S2D). Like the phenotypes observed in RPA experiments, FAM111A depletion reduced ssDNA exposure upon HU treatment (Fig 2I). Altogether, these results revealed that upon HU treatment, FAM111A promotes ssDNA formation.

ssDNA formation upon HU treatment arises from extensive DNA unwinding at arrested replisomes and activation of dormant origins (Ge et al, 2007; Mejlvang et al, 2014; Marchal et al, 2019). The defect in HU-induced ssDNA accumulation observed here in FAM111A-depleted cells may thus be caused by inefficient origin activation, as we have shown that FAM111A depletion impairs origin activation in response to replicative stress (Fig 1). Consistent with this, in FAM111A-depleted cells, ssDNA formation is impaired upon UCN-01 treatment (Fig S2E). We next tested the possibility that FAM111A plays a general role in promoting ssDNA exposure in S phase upon treatment to other genotoxic drugs. To this end, we monitored the ability of FAM111A to promote RPA accumulation upon treatment with the radio-mimetic agent bleomycin (Bleo) and the topoisomerase 1 inhibitor camptothecin (CPT), both drugs inducing an increased ssDNA formation due to repair at single- and double-strand breaks. As expected, upon CPT and Bleo treatments, RPA levels increase compared with untreated conditions (Fig S2F). In FAM111A-depleted cells, the RPA accumulation upon CPT and Bleo treatments was not impaired. Therefore, FAM111A does not have a general role of promoting ssDNA formation in response to genotoxic challenges. Instead, FAM111A promotes ssDNA formation upon fork stalling (HU and APH treatments). This novel function may be distinct from FAM111A role in overcoming protein–DNA complexes ahead of replisomes formed by topoisomerase 1 (Kojima et al, 2020) or PARP1 (Murai et al, 2012, 2014), as in either HU- or APH-treated cells, replisomes are not arrested because of the obstacles ahead of the fork.

## The paralogs FAM111A and FAM111B have only partially overlapping functions

Recent work has shown that in addition to a PIP domain in its N terminus, FAM111A also possesses a ssDNA-binding domain in its central regions (Fig 3A) (Kojima et al, 2020). Nevertheless, as

FAM111A substrate(s) remain unknown, it is unclear how FAM111A could facilitate origin activation and ssDNA formation. As a first strategy to identify putative substrates of FAM111A, we performed a FAM111A interactome analysis using affinity purification and mass spectrometry (AP-MS) of endogenous FAM111A from whole cell and chromatin extracts (Fig S3A and B). In both whole cells and chromatin extracts, FAM111A's top interactor was FAM111B (Fig 3B and C and Table S1). This interaction was confirmed by Western blotting (Fig S3C). PCNA and RFC-1, previously identified as FAM111A interactors upon FAM111A overexpression (Alabert et al, 2014; Hoffmann et al, 2020), were also identified by mass spectrometry but their enrichments remained non-significant (Table S1).

FAM111B is FAM111A paralog, and sequence alignment-based prediction suggests that like FAM111A, FAM111B contains a trypsin-like peptidase domain in the C-terminus (Fig S3D). A closer inspection of FAM111A and B sequence conservation revealed that the ssDNA-binding domain identified in FAM111A (Kojima et al, 2020) is well conserved in FAM111B (Figs 3D and S3E). Moreover, we identified in both paralogs two ubiquitin-like (UBL) repeat domains, U1 and U2, which differ from each other by the presence of a long positively charged loop rich in arginine and lysine between β-strands 1 and 2 in U2 (Fig 3E–I). Notably, the ssDNA-binding domain maps to the U2 domain (Figs 3A and S3D). Two other UBL domains are known to interact with nucleic acids, the SUMO-1 UBL which binds to double-stranded DNA (Eilebrecht et al, 2010) and the SF3A1 UBL domain which binds to double-stranded RNA (Martelly et al, 2019). To our knowledge, the FAM111A UBL domain is the first case of a putative UBL domain that interacts with ssDNA.

In addition to forming a complex, FAM111A and FAM111B are both transiently enriched on newly replicated chromatin (Fig 3J and [Alabert et al, 2014]). As FAM111B is FAM111A's top interactor, we tested whether FAM111B could be a substrate of FAM111A. To this end, FAM111A was overexpressed (OE) or depleted and FAM111B abundance examined by Western blot and quantitative mass spectrometry, anticipating that FAM111A's substrate abundance would vary under these conditions. Neither depletion nor overexpression of FAM111A affected FAM111B protein levels and vice versa (Figs 3K and S4A and B and Table S2), suggesting that although FAM111A and FAM111B may form a complex, they unlikely cleave one another. FAM111B has been suggested to promote cell cycle progression through the degradation of the cell cycle inhibitor p16 (Kawasaki et al, 2020). As a FAM111B paralog, FAM111A could promote S phase entry by a similar mechanism, targeting p16 or another cell cycle inhibitor. Yet, FAM111A OE did not affect the level of p16 or other cell cycle inhibitors such as p21 (Fig S4B and Table S2). Based on findings from Figs 1 and 2, another possibility is that FAM111A targets a kinase, or a phosphatase, involved in origin activation. Phosphoproteomic analysis revealed that a handful of proteins

---

labeling. Red bar represents the median; n > 100 inter CldU were analyzed. **(G)** Chromatin-bound MCM2 levels in U-2-OS cells shown as a function of DAPI intensity and cell cycle stage detected by QIBC. EdU-based gating strategy shown in Fig S1C. Green, G1 phase; blue, early S phase; red, mid/late S phase; grey, G2/M phase. From left, n = 2,317, 2,335, 1,424. **(G, H)** Quantification of chromatin-bound MCM2 in G1 phase analyzed in (G). From left, n = 474, 999, 380. **(I)** Chromatin binding of Cdc45 in S phase cells detected by QIBC. Cdc45 levels are shown as a function of DAPI intensity, S phase cells were gated based on chromatin-bound proliferating cell nuclear antigen intensities (Fig S1I). **(I, J)** Quantification of chromatin-bound Cdc45 in S phase cells analyzed in (I). From left, n = 1,329, 1,103, 1,155. **(A, B, C, D, E, F, G, H, I, J)** Data are representative of two (B, E, F) and three (A, C, D, G, H, I, J) independent experiments. siControl, non-targeting siRNA; a.u., arbitrary units. **(B)**, unpaired t test. **(E, F, H, J)**, Mann–Whitney test. ***P < 0.001, **P < 0.01, *P < 0.05.
Source data are available for this figure.

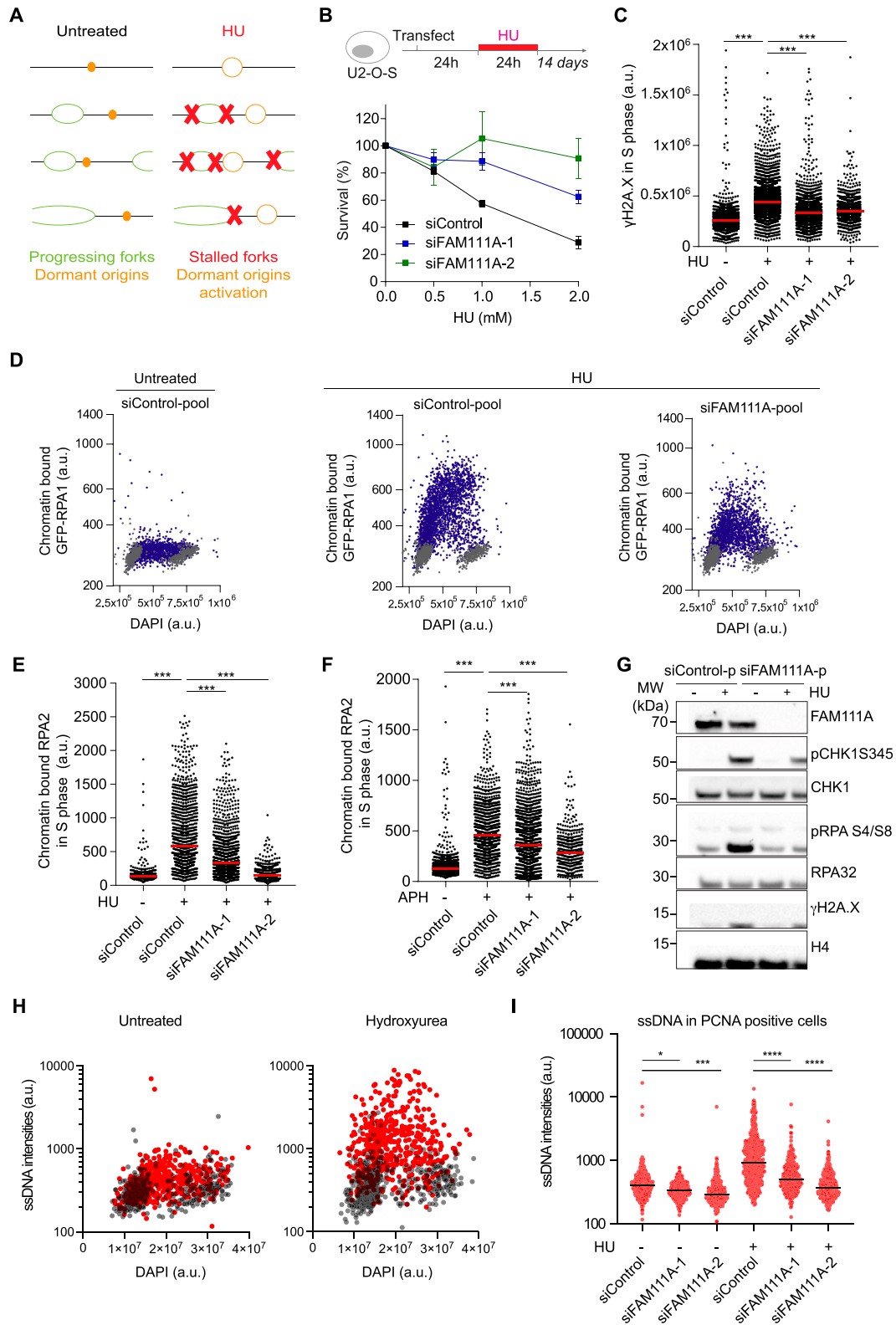

**Figure 2. FAM111A depletion protects cells against replicative stress.**
**(A)** Schematic representation of active forks and dormant origins under unchallenged conditions and upon HU treatment. In response to HU, ongoing forks stall and dormant origins fire, leading to increased amount of single-stranded DNA (ssDNA) exposed and subsequent RPA loading. **(B)** Clonogenic survival assays of siRNA-transfected cells treated with HU as indicated. **(C)** Chromatin abundance of γH2A.X in S phase analyzed as in Fig S1K. From left, n = 2,720, 2,486, 1,664, 1,149. **(D)** Chromatin-bound RPA1-GFP intensity in U-2-OS cells treated with 3 mM HU for 2 h and detected by Quantitative Image-Based Cytometry. Data shown as a function of DAPI intensity.

linked to DNA replication such as RIF1 and ORCA showed increased phosphorylation at defined sites upon FAM111A depletion (Fig S4C and Table S3). Although RIF1 has been shown to negatively regulate origin activation through opposing DDK functions, no studies to date have implicated S1579 phosphorylation (identified here) in regulating this function (Moiseeva et al, 2019). Similarly, S243 phosphorylation of ORCA has only been described in large-scale proteomics studies and has no known described roles (Sahu et al, 2023). Altogether, these results suggest that FAM111A does not target FAM111B for degradation, and functional FAM111A's targets remain to be identified.

To better understand the relationship between the two paralogs, individual and combined effects of FAM111A and FAM111B depletion and overexpression were directly compared. Single depletion of either FAM111A or FAM111B reduced DNA synthesis rates, whereas co-depletion of FAM111A and FAM111B did not promote a further reduction of DNA synthesis (Figs 3L and S4B). This was also true for RPA reduction (Fig S4D), suggesting that the two paralogs may be epistatic. Consistent with earlier findings (Hoffmann et al, 2020), FAM111A OE induces DNA damage, whereas FAM111B OE does not (Fig 3K), suggesting that FAM111A has FAM111B independent function(s). Moreover, simultaneous OE of FAM111A and FAM111B mirrors the effect of FAM111A OE alone (Fig S4E), revealing that the toxicity of FAM111A OE is not due to changes in the ratio of the two proteins under these conditions. Altogether, these data revealed that although FAM111A and FAM111B form a complex and promote DNA replication, they may have only partially overlapping functions. Consistent with this, mutations in FAM111A and FAM111B are associated with a distinct set of diseases (Welter & Machida, 2022).

### Unrestrained FAM111A peptidase activity leads to extensive ssDNA formation in replicating cells

The two KCS2 patient FAM111A mutations R569H and Y511H are dominant and predicted to confer hyperactive peptidase activity (Kojima et al, 2020). Based on our experiments, we hypothesized that part of the deleterious phenotype observed in KCS2 patients could be due to excessive ssDNA formation. To test this possibility, stable cell lines conditionally expressing either WT or mutant FLAG-HA-FAM111A mutants were generated (Fig S5A). The PIP mutation disrupts the direct binding of FAM111A to PCNA (Alabert et al, 2014), the S541A mutation generates a FAM111A putative peptidase dead mutant and the R569H, Y511H, and T338A mutations potentiate FAM111A peptidase activity (Hoffmann et al, 2020; Kojima et al, 2020). FAM111A WT, PIP mutant and diseases mutants' expression increased γH2A.X levels and promoted cell death (Figs 4A and S5B and C). In contrast, expression of the peptidase dead mutant S541A did not increase γH2AX levels (Figs 4B and S5D) or caused cell death (Fig S5C). Consistent with this, when R569H, the most frequent disease

mutation, was combined with the S541A peptidase dead mutation, γH2A.X levels were rescued (Fig 4B). The increased γH2AX levels observed upon FAM111A WT OE can also be suppressed by treating cells simultaneously with the pan caspase inhibitor Z-VAD-FMK (Fig 4C). This is consistent with earlier findings (Hoffmann et al, 2020) and confirms that unrestrained FAM111A peptidase activity leads to DNA damage formation and is deleterious to cell survival. Furthermore, it confirms that the increased γH2AX is most likely a consequence and not a cause of FAM111A-induced apoptosis.

FAM111A OE has been shown to lead to defective DNA replication but the cause of this remains unclear. Based on our findings, we wanted to test whether ssDNA accumulation could be one of the primary cellular stresses caused by FAM111A OE. We thus overexpressed FAM111A in presence of the pan caspase inhibitor and monitored EdU, RPA, and ssDNA formation (Fig S5E). We found that the DNA synthesis defects were not suppressed by the pan caspase inhibitor treatment (Fig 4), supporting that FAM111A OE arrests DNA synthesis independently of apoptosis. There was also an anti-correlation between EdU levels and FAM111A protein levels (Fig S5F). Notably, a fraction of cells with low EdU signal showed an accumulation of chromatin-bound RPA which was not suppressed by the caspase inhibitor treatment (Fig 4D). Consistent with these observations, following FAM111A OE, ssDNA levels also increased and could not be rescued through caspase inhibition (Fig 4E). Importantly, OE of the peptidase dead mutant version of FAM111A did not affect ssDNA level (Fig 4F). Altogether, these data revealed that unrestrained FAM111A peptidase activity leads to ssDNA formation independently of apoptosis. On the other hand, unlike FAM111A, FAM111B OE did not promote DNA damage (Fig 3J) or ssDNA formation (Fig 4G).

Upon FAM111A OE, cells were arrested at the G1/S transition (Fig S5G) and two populations of RPA-positive cells were detected, EdU positive and EdU negative (Fig 4D), suggesting that FAM111A may promote ssDNA formation in G1 phase cells. Mechanistically, it would suggest that FAM111A can promote ssDNA formation independently of PCNA and outside of S phase. To test these two possibilities, we first measured ssDNA formation upon OE of FAM111A PIPmt. OE of FAM111A PIPmt lead to ssDNA formation but to a smaller extent compared with FAM111A WT (Figs 4H and S5H), revealing that FAM111A binding to PCNA only partially contributes to ssDNA formation. Next, we used different cell cycle inhibitors to block cells in G1 phase (MCM loading, PD0332991; Origin activation, TAK-931), at the G1/S phase transition (Thymidine) or allow cells to enter S phase (Untreated, Bleo), and monitored ssDNA formation upon FAM111A OE (Figs 4I and S5I). Blocking cells in G1 phase (PD0332991 or TAK-931) or at the G1/S transition (Thy) significantly reduced FAM111A induced ssDNA formation (Fig 4J). Collectively, these results reveal that ssDNA

---

Gating strategy as in Fig S1K. Blue, proliferating cell nuclear antigen (PCNA) positive; grey, PCNA negative. **(E)** Quantification of chromatin-bound RPA2 in S phase cells treated with 3 mM HU for 2 h, analyzed as in Fig S1I. **(F)** Chromatin binding of RPA2 in S phase cells treated with 50 µg/ml aphidicolin for 2 h, analyzed as in Fig S1K. From left, n = 1,494, 999, 1,349, 402. **(G)** Immunoblot of whole cell extracts from siRNA-transfected cells treated as in Fig S2A. **(H)** ssDNA intensity in U-2-OS cells treated with 3 mM HU for 2 h and detected by Quantitative Image-Based Cytometry. Data shown as a function of DAPI intensity. Gating strategy as in Fig S1I. Red, PCNA positive; grey, PCNA negative. **(I)** Quantification of ssDNA in PCNA-positive cells treated with 3 mM HU for 2 h. From left, n = 521, 225, 451, 654, 251, 552. **(B, C, D, E, F, G, H, I)** Data are representative of three (C, D, E, F, G, H, I) and two (B) independent experiments. **(C, E, F, I)**, Mann–Whitney test, ***P < 0.001.
Source data are available for this figure.

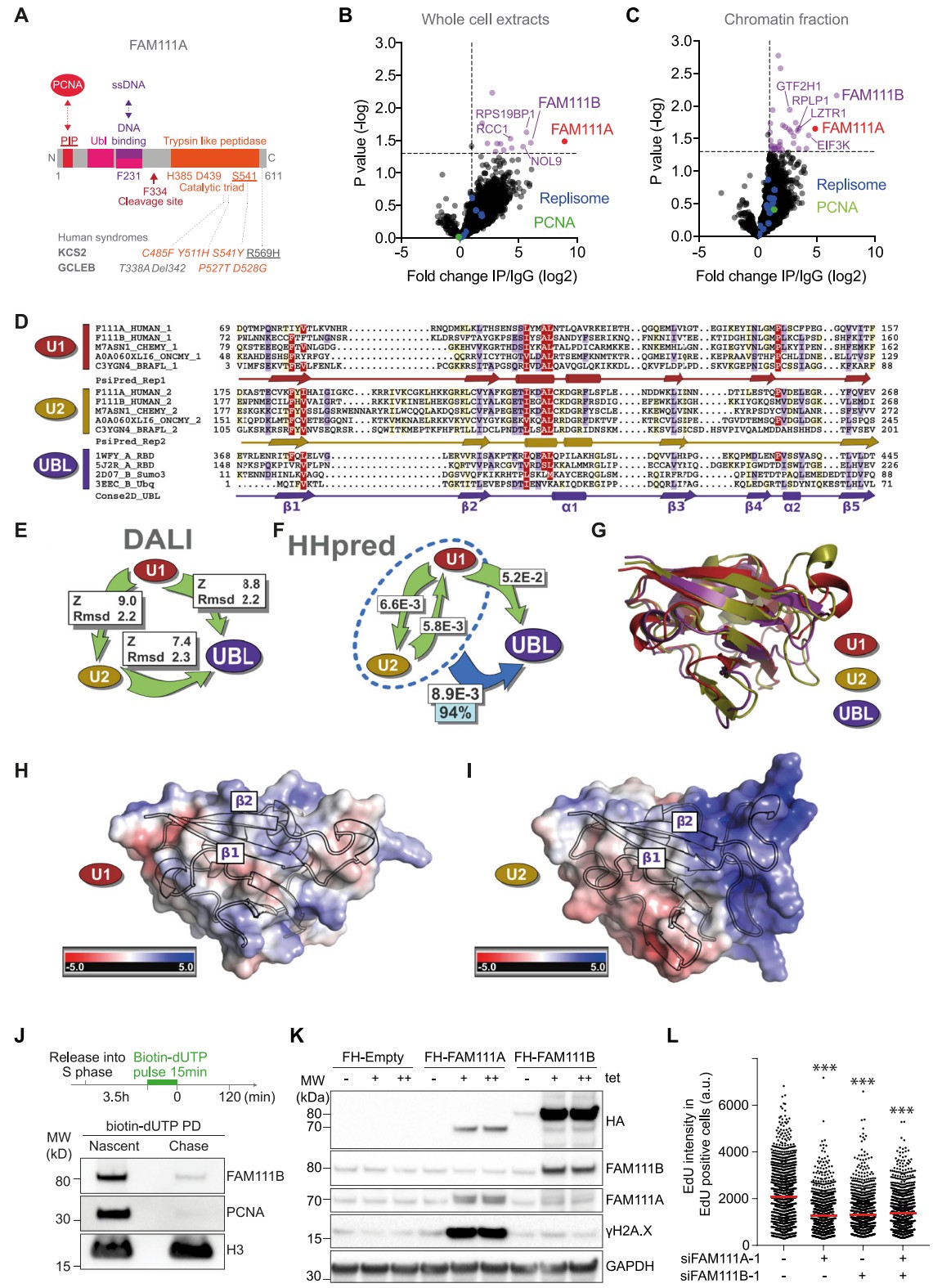

**Figure 3. Identification of FAM111A binding partners.**
**(A)** Schematic representation of FAM111A domain structure with notable residues and direct interactors highlighted. **(B, C)** FAM111A complexes from whole cell extract ((B), n = 3) and chromatin fraction ((C), n = 3). **(D)** Multiple sequence alignment of two consecutive ubiquitin-like (UBL) domains in FAM111. Red, FAM111 UBL repeats 1 (U1); yellow, FAM111 UBL repeats 2 (U2); purple, selection of UBL domains with known structure (UBL). Secondary structure predictions were performed independently for U1 (PsiPred_Rep1 lane) and U2 (PsiPred_Rep2 lanes) and are consistent with UBL (Conse2D_UBL lane). α-helices, cylinders; β-strands, arrows. Average BLOSUM62 score:

formation upon FAM111A OE relies on an intact FAM111A peptidase domain and on S phase entry.

## Discussion

The autocleavage site of FAM111A suggests a chymotrypsin-like peptidase specificity (Hoffmann et al, 2020; Kojima et al, 2020). Predicting its substrates in silico is unlikely as protease substrate specificities are often broad and highly dependent on amino acid sequence and tertiary structure (Goettig et al, 2019). To our knowledge, the only known FAM111A substrates are FAM111A itself, and upon CPT and PARPi treatment, TOP1 and PARP, respectively (Murai et al, 2014; Kojima et al, 2020). We found that FAM111B is FAM111A's top interactor but is unlikely its substrate and vice versa. Moreover, our data revealed that FAM111A and FAM111B act epistatically to ensure efficient DNA replication. Interestingly, it has been recently shown that FAM111B degrades p16, a cell cycle inhibitor of G1/S entry. Whether FAM111A can also target cell cycle inhibitors and whether the epistasis between FAM111A and FAM111B translates to this and other aspects of FAM111 biology remains to be defined.

Our data reveal that FAM111A depletion impairs origin activation and ssDNA formation. Both functions could be achieved indirectly by degrading a protein preventing S phase entry, as both mechanisms rely on S phase entry. Alternatively, FAM111A may directly degrade an essential protein, leading to fork stalling (Hoffmann et al, 2020) or degrade a DNA-binding protein(s) blocking origin activation and ssDNA exposure. The latter is reminiscent of FAM111A role in degrading DNA bound TOP1 and PARP under condition of replicative stress. Moreover, it provides a rationale for the deleteriousness of FAM111A gain of function mutations in KCS2 patients. Indeed, haploinsufficiency is unlikely to explain the pathogenic mechanism in KCS and OCS patients (Welter & Machida, 2022). Instead, FAM111A disease mutations such as R569H have been shown to be gain of function mutation, with FAM111A becoming constitutively active (Hoffmann et al, 2020; Kojima et al, 2020). In this study, we provide evidence of an equilibrium between FAM111A peptidase activity and ssDNA formation (Fig 4K). We show that ssDNA formation requires S phase entry, FAM111A protease activity, and occurs independently of apoptotic signaling. Whether extensive ssDNA formation directly contributes to the deleterious effects seen in patients remains to be tested. Overall, our data highlight how FAM111A may play positive roles in DNA replication under basal conditions although becoming harmful upon unrestrained expression of peptidase domain and patient mutations. Developing FAM111A peptidase domain inhibitors may thus be beneficial for our understanding of KCS2 and GCLEB syndrome's etiology.

## Materials and Methods

### Cell lines and cell culture conditions

U-2-OS (ATCC), HeLa S3 (ATCC), GFP-RPA1, and RFP-PCNA U-2-OS and Flp-In T-Rex U-2-OS cells were grown in DMEM (Gibco) containing 10% FBS, 1% penicillin/streptomycin, and drugs for selection. Flag-HA-FAM111A WT (WT) and Flag-HA-FAM111B plasmids were generated in the pcDNA5/FRT/TO backbone. Flag-HA-Y24A-Y25A (PIPmt), Flag-HA-S541A, Flag-HA-R569H, Flag-HA-R569H-S541A, Flag-HA-Y511H, and Flag-HA-T338A plasmids were generated from Flag-HA-FAM111A WT construct by site-directed mutagenesis. All plasmids were confirmed by Sanger sequencing. Cells inducible for FLAG-HA-FAM111A mutants and Flag-HA-FAM111B were generated in Flp-In T-REx U-2-OS cells by transfection of the above constructs with Lipofectamine 2000 (Invitrogen) according to the manufacturer's protocol, and selection with 100 μg/ml hygromycin.

### Transfections and siRNA

siRNAs were introduced by Lipofectamine RNAiMAX (Invitrogen), according to manufacturer's recommendations. The following siRNAs were used. siFAM111A-1, GUAAUCAGUUUCAUGACACUAA-AdAdG and siFAM111A-2, ACCUUGGUUUGAGAUACAUAAUGdAdA (SR324823; Origene); siFAM111A-p (pool), GCAUUGUGGGAGACGGAAU, UACUGAAACUGUCGGAAUA, CGAUUAAAGUAGUGAAACU, GGUCAAUGU-GUAAGGGUGA (ON-TARGETplus SMART human FAM1111A [63091], L-013926-01-0005; Dharmacon]; siFAM111B-1: GCUUAAAGUGUCCAAU-GAAAACTA (SR317776; Origene). Control siRNAs: siControl, CGUU-AAUCGCGUAUAAUACGCGUdAdT (SR30004; Origene), siControl-p (pool), UGGUUUACAUGUCGACUAA, UGGUUUACAUGUUGUGUGA, UGGUUUACAUGUUUUCUGA, UGGUUUACAUGUUUUCCUA (ON-TARGETplus Non-targeting Pool, D-001810-10-15; Dharmacon). For transient transfection of FAM111A, plasmid FAM111A-turboGFP (RG210012; Origene) was transfected with Lipofectamine 2000 (11668; Invitrogen) following supplier guidelines (10 μl Lipofectamine and 2.5 μg of plasmid for one well of a six-well plate). 6 h later, medium is changed.

red, >1.5; violet, between 1.5 and 0.5; light yellow, between 0.5 and 0.2. **(E)** Structural comparison of the UBL FAM111A U1 and U2 AlphaFold models, Z-Scores, and root mean square deviation arising from Dali structural superpositions and estimates their structural similarity. Root mean square deviation is the average distance (in angstroms) between the backbone atoms of superimposed proteins. **(F)** HHpred analysis. White rectangles, HHpred profile-versus-profile comparison $E$-values from global profile search results. Arrows, profile search direction, for example, U1 aligns to U2 with $E$-value = 6.6 × 10$^{-3}$. Dotted blue oval, HHpred searches against the PDB70 profile database using alignment of U1 and U2 repeats as input detected the UBL Ras-binding domain of mouse RGS14 (PDB ID: 1WFY) (UBL) with $E$-value of 8.9 × 10$^{-3}$; cyan rectangle, true-positive homology probability of 94%. **(G)** Structural superposition of UBL FAM111A U1 and U2 AlphaFold models (in red and yellow, respectively) and the UBL Ras-binding domain of mouse RGS14 (PDB ID: 1WFY) (in purple). **(H, I)** AlphaFold 3D models of FAM111A U1 and U2 repeats. Red, negative charge surface electrostatic potential; blue positive. **(J)** NCC analysis of FAM111B recruitment to nascent chromatin in HeLa S3 cells. **(K)** Immunoblot of whole cell extracts after induction of Flag-HA-FAM111A or Flag-HA-FAM111B. FH- Flag-HA; (+), 0.5 μg/ml tet; (++), 1 μg/ml tet. **(L)** EdU intensity in EdU-positive cells in siRNA-transfected U-2-OS cells. n = 1,307, 974, 909, 986. **(B, C, J, K, L)** Data are representative of three (B, C, J, K) and two (L) independent experiments. **(L)**, Mann–Whitney test, ***$P$ < 0.001. Source data are available for this figure.

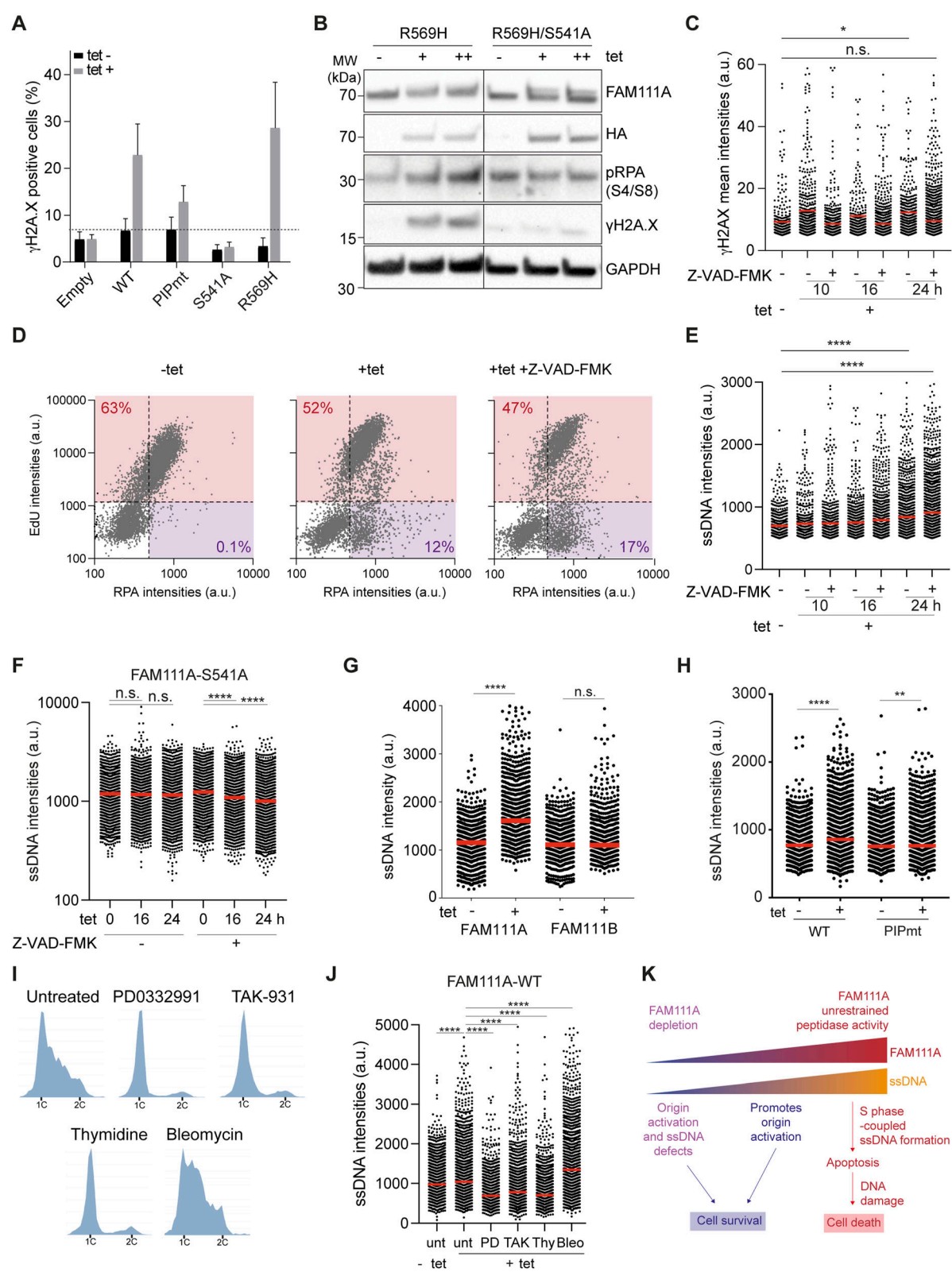

**Figure 4. Unrestrained FAM111A peptidase activity promotes single-stranded DNA (ssDNA) exposure.**
**(A)** Quantification of γH2AX-positive cells upon Flag-HA-FAM111A overexpression. **(B)** Immunoblot of whole cell extracts from asynchronous cells 24 h after tetracycline induction. **(C)** Quantification of γH2AX intensities upon FAM111A overexpression by Quantitative Image-Based Cytometry (QIBC). From left, n = 733, 913, 1,220, 953, 986, 1,315, 1,925. **(D)** EdU and RPA intensities per cell in U-2-OS cells upon Flag-HA-FAM111A overexpression, detected by QIBC. Percentage of EdU-positive cells (red) or Edu negative, RPA positive (purple) are shown. **(E)** Quantification of ssDNA upon Flag-HA-FAM111A overexpression. **(F)** Quantification of ssDNA upon FAM111A-S541 (peptidase dead)

## Drug treatments

Hydroxyurea (3 mM for 2 h for QIBC and immunoblotting, 0.5 mM, 1 mM, or 2 mM for 24 h for clonogenic assays, H8627; Sigma-Aldrich), UCN-01 (300 nM for 1 h, U6508; Sigma-Aldrich), APH (50 µg/ml for 2 h, A4487; Sigma-Aldrich), thymidine (2.2 mM for 12 or 17 h, T1895; Sigma-Aldrich), nocodazole (100 ng/ml for 12 h, M1404; Sigma-Aldrich), PD0332991 (5 µM for 12 h, PZ0383; Sigma-Aldrich), TAK-931 (300 nM for 12 h, HY-10088; Biotech), bleomycin (25 µg/µl for 3 h on and 3 h off, S1214; Stratech), CPT (500 nM for 6 h, C9911; Sigma-Aldrich), Z-VAD-FMK (50 µM, ab120487; Abcam). Protein expression was induced in with 0.5–1 µg/ml tetracycline for 24 h (T7660; Sigma-Aldrich).

## Sub-cellular fractionations

Cytoplasmic, nuclear, and chromatin-bound fractions were isolated as previously described (Mendez & Stillman, 2000). Briefly, to isolate chromatin, cells were resuspended (4 × 10$^7$ cells/ml) in 0.1% Triton X-100 Buffer A (10 mM Hepes pH 7.9, 10 mM KCl, 1.5 mM MgCl$_2$, 0.34 M sucrose, 10% glycerol, 1 mM DTT, protease inhibitors) for 8 min on ice. Nuclei were pelleted at 1,300$g$, 4°C for 4 min (P1). The cytosolic supernatant (S1) was further clarified for 15 min at 20,000$g$, 4°C. P1 was washed in buffer A and incubated with buffer B (3 mM EDTA, 0.2 mM EGTA, 1 mM DTT, protease inhibitors) for 30 min on ice, centrifuged (4 min, 1,700$g$, 4°C), washed with buffer B, and centrifuged again. For immunoprecipitations, the chromatin pellet (P3) was further resuspended in buffer C (20 mM Hepes pH 7.9, 1.5 mM MgCl$_2$, 420 mM NaCl, 25% glycerol, 1 mM DTT, protease inhibitors) and incubated with benzonase (70746-3; Millipore) on ice for 30 min. For immunoprecipitation from whole cell extracts, cells were lysed with 420 mM NaCl, 20% glycerol, 10 mM Hepes, pH 7.9, 0.1% NP-40, 1 mM EDTA, 1 mM DTT, protease inhibitors). Samples were then syringed with a 25G syringe 10 times and benzonase treated on ice for 30 min. To analyze soluble and chromatin-bound fractions by immunoblotting, cells were treated as previously described (Saredi et al, 2016). Briefly, cells were incubated in 0.5% Triton X-100 CSK buffer (10 mM PIPES, pH 7, 100 mM NaCl, 300 mM sucrose, 3 mM MgCl$_2$), supplemented with protease and phosphatase inhibitors (5 mM sodium fluoride, 10 mM β-glycerolphosphate, 0.2 mM sodium vanadate, 10 µg/ml leupeptin, 10 µg/pepstatin, 0.1 mM PMSF) on ice for 10 min and centrifuged at 1,500$g$ for 10 min to collect soluble proteins. Pellets were washed again in CSK buffer, resuspended with SDS lysis buffer (1% SDS, 50 mM Tris–HCl, pH 8.1, 10 mM EDTA, 1 mM PMSF, 10 mM MgCl$_2$, protease inhibitors) and treated with benzonase for 30 min. For immunoblotting of whole cell extracts, cells were lysed in SDS lysis buffer as above.

## Nascent chromatin capture

NCC was performed as described previously (Alabert et al, 2014). Cells were synchronized by single thymidine block (2.2 mM) for 17 h and released into S phase with 24 µM 2'deoxycytidine for 3.5 h. Cells were labeled with 50 µM biotin-dUTP for 5 min in hypotonic buffer (50 mM KCl, 10 mM Hepes), supplemented with medium for 15 min, chased with biotin-dUTP–free medium for indicated times and fixed for 15 min in 2% formaldehyde. Nuclei were isolated by douncing 20 times in sucrose buffer (0.3 M sucrose, 10 mM Hepes, pH 7.9, 1% Triton X-100, 2 mM MgOAc). Chromatin was solubilized in sonication buffer (10 mM Hepes, pH 7.9, 100 mM NaCl, 2 mM EDTA, 1 mM EGTA, 0.2% SDS, 0.1% sodium sarkosyl, 1 mM PMSF) using Diagenode Bioruptor (28 cycles, 30 s on, 90 s off, high intensity). Biotin-dUTP–labeled chromatin was purified on Streptavidin C1 Dyna-beads (Invitrogen) overnight. Isolated nascent chromatin was boiled for 40 min in LSB (50 mM Tris–HCl at pH 6.8, 100 mM DTT, 2% SDS, 10% glycerol, bromophenol blue).

## Quantitative image–based cytometry

U-2-OS cells were grown on clear bottom 96-well plates (Greiner) and either pre-extracted with cold 0.5% Triton X-100 CSK buffer for 5 min before fixation or directly fixed in 4% formaldehyde for 10 min. For EdU (5-ethynyl-2'-deoxyuridine) labeling, cells were incubated with 40 µM EdU for 20–30 min. EdU was detected using Click-iT Plus EdU Kit for Imaging (C10640). Plates were imaged on a Perkin Elmer Operetta high-content imaging system or Olympus Scan-R imaging system using a 20x objective. 35 fields per well were imaged, and ~2,000 cells per condition were analyzed. Single-cell fluorophore intensities were extracted using the Columbus system (Perkin Elmer) or Scan-R analysis software. Cell cycle phases were gated based on DAPI and EdU or PCNA intensities. Graphs were generated using Tableau 2019.3 and GraphPad Prism.9 software.

## Clonogenic assay

U-2-OS were transfected with siRNAs and after 24 h seeded in technical triplicates or duplicates of 2,000 and/or 4,000 cells in 10 cm dishes. 48 h after transfection, treatments were performed as indicated. Cells were then cultured in fresh medium for 10–14 d, fixed in methanol and stained with 25% Giemsa stain in methanol. Colony formation efficiency was determined blindly by manual colony counting and normalized to untreated controls. Each data

overexpression. n > 650 per condition. **(G)** Quantification of ssDNA upon FAM111A and B overexpression. n > 478 per condition. **(H)** Quantification of ssDNA upon FAM111A PIPmt overexpression. n > 1969 per condition. **(I)** Cell cycle distribution based on DAPI content (QIBC) of cells upon Flag-HA FAM111A WT overexpression and subsequent treatment with genotoxic agents, bleomycin (Bleo) and HU, or cell cycle inhibitors, PD0332991 (PD), TAK-931 (TAK), or thymidine (Thy). Experimental outline in Fig S5I. **(J)** Quantification of ssDNA upon different treatments From left, n = 2,345, 3,456, 2,567, 5,889, 4,367, 3,992, 5,896. **(K)** Model of FAM111A function. FAM111A plays a positive role in DNA replication, promoting origin firing and ssDNA formation. In a disease context, unrestrained peptidase activity causes extensive ssDNA formation and apoptosis. **(A, B, C, D, E, F, G, H, I, J)** Data are representative of three (A, C, D, E, F) and two (B, G, H, I, J) independent experiments. For (A), data are represented as mean + SD of n = 3 experiments. tet (−), uninduced cells; tet (+), 0.5 µg/ml tetracycline; tet (++), 1.0 µg/ml tetracycline. (C, E, F, G, H, J) Red bars, mean; unpaired $t$ test, ****$P$ < 0.0001, *$P$ < 0.05, n.s., non-significant.
Source data are available for this figure.

point represents a technical replicate of 2,000 cells seeded cells within each biological replicate.

## Cell proliferation assay

U-2-OS were transfected with siRNAs and after 24 h seeded in clear bottom 96-well plates (Greiner) at 500 cells/well in triplicates. Cell viability was measured every 24 h for 4 d using CellTiter-Glo 2.0 Cell Viability Assay (G9242; Promega) according to manufacturer's recommendations.

## Molecular DNA combing

Single-molecule analysis of DNA replication by molecular combing was performed as described in protocol 36 available from the EpiGeneSys Network of Excellence website. In brief, 48 h after siRNA transfections, U-2-OS cells were successively pulse labeled with 50 $\mu$M CldU and 250 $\mu$M IdU for 30 min each. Immediately after the pulse, cells were harvested, molded into low-melting agarose plugs, and DNA was extracted using Fiber Prep DNA extraction kit (EXT-002; Genomic Vision). DNA was combed on silanized coverslips (Genomic Vision) using FiberComb (Molecular Combing System; Genomic Vision)-combed coverslips were subsequently dehydrated at 60°C for 2 h, denatured in denaturing solution (0.5 M NaOH, 1 M NaCl) for 8 min, blocked with Block Aid solution (Invitrogen) for 30 min at 37°C, and probed with rat anti-BrdU (ab6326, 1/25; Abcam) and mouse anti-BrdU (347580, 2/25; BD Biosciences) antibodies for 1 h, and anti-ssDNA (MAB3034, 4/25; Merck Millipore) for 2 h. Coverslips were then washed with 0.05% PBST and incubated with anti-mouse Cy3.5-conjugated IgG (ab6946, 1/25; Abcam) and anti-rat Cy5-conjugated IgG (ab6565, 1/25; Abcam), and washed again. Immunolabeled coverslips were then dehydrated and sent for automated imaging using Genomic Vision EasyScan service. More than 300 fibers were analyzed for each condition. Measured distances were converted to kilobases by the constant stretching factor (1 $\mu$m = 2 kb). Inter-fork distances were determined based on CldU staining.

## Immunofluorescence-based detection of ssDNA

Experiments were performed as described previously (Mejlvang et al, 2014). Cells were pulse labeled with 10 $\mu$M BrdU (Cat# B5002; Sigma-Aldrich) for 24 h after siRNA transfection. ssDNA was subsequently revealed by BrdU detection under non-denaturing conditions (the BrdU epitope is not detected by anti-BrdU antibodies in double-strand DNA). For detection of total BrdU incorporation in double-strand DNA, fixed cells were treated with 4 M HCl (10 min) to denature DNA before immunostaining.

## Flow cytometry

For cell cycle analysis, cells were trypsinized, fixed with ice-cold 70% ethanol overnight at 4°C, treated with propidium iodide (PI) solution (50 $\mu$g/ml PI, 50 $\mu$g/ml RNaseA, and 0.1% Triton X-100, 1% FBS in PBS) for 30 min, and acquired using BD FACSCanto. Results were analyzed using FlowJo software.

## Immunoprecipitation from cell extracts

Immunoprecipitations of endogenous FAM111A were performed with anti-FAM111A (ab184572; Abcam) or control IgG bound to protein A Dynabeads (10002D; Thermo Fisher Scientific) and incubated rotating overnight at 4°C. Beads were washed three times with wash buffer (20 mM Hepes, pH 7.9, 150 mM KCl, 2.5 mM MgCl$_2$, 1 mM DTT, 0.5 mM PMSF, 0.1% NP-40), boiled in LSB, and subjected to SDS–PAGE separation on NuPAGE 4–12% gels (NP0321; Invitrogen).

## AP-MS analysis

Immunoprecipitated fractions were analyzed as follows. Samples were run on 4–12% NuPAGE gels (#NP0321; Invitrogen) until top and bottom markers were separated by 1 cm. Gel slices were excised, broken into small pieces, successively washed for 15 min with 100 mM ammonium bicarbonate, 100 mM ammonium bicarbonate/acetonitrile (50:50) and acetonitrile, and dried in vacuo. Samples were then incubated in reducing solution (10 mM DTT, 20 mM ammonium bicarbonate) for 60 min at 56°C, alkylated with 50 mM iodoacetamide in 20 mM ammonium bicarbonate for 30 min at RT, washed twice with 100 mM ammonium bicarbonate for 15 min and with acetonitrile for 10 min, and then dried in vacuo. Digestion was performed with trypsin (12.5 $\mu$g/ml) overnight at 30°C. Samples were then extracted with consecutive acetonitrile washes, pooled, and dried in vacuo. Label-free peptide analysis was performed by the FingerPrints Proteomics Facility (University of Dundee) on a Q Exactive plus mass spectrometer (Thermo Fisher Scientific) coupled with a Dionex Ultimate 3000 Rapid Separation LC (Thermo Fisher Scientific). The following LC buffers were used: buffer A (0.1% formic acid in Milli-Q water [vol/vol]) and buffer B (80% acetonitrile and 0.1% formic acid in Milli-Q water [vol/vol]). Aliquots of 10 $\mu$l were loaded at 10 $\mu$l/min onto a trap column (100 $\mu$m × 2 cm; PepMap nanoViper C18 column, 5 $\mu$m, 100 Å; Thermo Fisher Scientific) equilibrated with 98% buffer A. The trap column was washed for 5 min at the same flow rate and then switched in-line with PepMap RSLC C18 column (75 $\mu$m × 50 cm, 2 $\mu$m, 100 Å). The peptides were eluted from the column at 300 nl/min with a linear gradient of 2–35% buffer B over 125 min and then 98% buffer B by 127 min. The column was then washed with 98% buffer B for 20 min and re-equilibrated in 2% buffer B for 17 min. Q-Exactive plus was operated in positive mode using data-dependent mode. A scan cycle comprised MS1 scan (m/z range from 330–1,600, with a maximum ion injection time of 20 ms, a resolution of 70,000, and automatic gain control [AGC] value of 1 × 10$^6$ followed by 15 sequential-dependent MS2 scans [with an isolation window set to 1.4 kD, resolution at 17,500, maximum ion injection time at 100 ms, and AGC 2 × 10$^5$]). Dynamic exclusion was set at 45 s, stepped collision energy was set to 27 and fixed first mass to 100 m/z. Spectrum was acquired in centroid mode and unassigned charge states, charge states above six and singly charged species were rejected. The raw AP-MS data were searched using MaxQuant (1.6.6.0) (Cox & Mann, 2008) and the Andromeda search engine software (Cox et al, 2011), and searched against Homo sapiens database from UniProt (December 2019; SwissProt). Data were searched with the following parameters: variable modification of oxidation (M), deamidation (N, Q), and acetylation (protein N terminus) and fixed modification of

carbamidomethylation (C). MS/MS tolerance: FTMS was set at 10 ppm and ITMS at 0.06 kD. The FDR threshold was set to 1%, allowing for maximum peptide length of eight and two missed cleavages. The proteinGroups output table from MaxQuant was analyzed using Perseus (1.6.7.0) (Tyanova et al, 2016a). Data were filtered to remove entries matched to "potential contaminant," "only identified by site," and "reverse." Entries with less than two unique peptides and not present in at least two FAM111A pull downs were also removed. Missing values were imputed from a normal distribution using default settings. Log$_2$ ratios were compared using two sample $t$ test.

## TMT MS analysis

Cells were washed with ice-cold PBS three times, resuspended in urea lysis buffer (8 M urea in 100 mM Tris–HCl, pH 8, 1x Complete Mini EDTA-free Protease Inhibitor Cocktail), mixed for 15 min at RT, and sonicated (Diagenode Bioruptor) for 30 cycles (30 s ON and 30 s OFF, high). The cell extracts were then cleared at 15,000$g$ for 10 min, and supernatants were treated with 10 mM tris (2-carboxyethyl) phosphine (TCEP) at RT for 45 min, followed by 20 mM iodoacetamide at RT in the dark for 30 min. Protein samples were further processed using SP3 magnetic beads (GE Healthcare Life Sciences) to remove all the salts/contaminants and digested to peptides for TMT labeling (Hughes et al, 2019). In detail, samples were mixed with SP3 beads (1:10) in 70% acetonitrile and incubated for 10 min at RT. Beads were washed twice with 1 ml 70% ethanol, once with 1 ml acetonitrile, re-dissolved in 80 $\mu$l 50 mM ammonium bicarbonate, and digested with trypsin (1: 50; trypsin: protein) at 37°C overnight. Samples were acidified by adding 9 $\mu$l 10% formic acid and acetonitrile to 95%, incubating 10 min at RT. Beads were washed with acetonitrile and re-dissolved in 2% DMSO. Equal amounts (100 $\mu$g) of each peptide sample were dried and dissolved in 100 $\mu$l 100 mM TEAB buffer. TMT labeling of each sample were followed by the TMT10plex Isobaric Mass Tag Labeling Kit (Thermo Fisher Scientific) manuals. TMT-labeled peptides were fractionated using off-line high-pH RP chromatography. The samples were loaded onto XBridge BEH C18 column (130 Å, 3.5 $\mu$m, 4.6 × 250 mm; Waters) and separated on Dionex BioRS HPLC system. The gradient was as follows: solvents A (water), B (acetonitrile), and C (100 mM ammonium formate, pH 9); 0–8 min, 5% B; 8–10 min, 5–21.5% B; 10–21 min, 21.5–48.8% B; 21–22 min, 48.8–90% B, 22–27 min, 90% B; 1 ml/min flow rate, with 10% C all through the gradient. Peptides were separated into 48 fractions, which were collected into 24 fractions. Fractions were subsequently dried and re-dissolved in 5% formic acid. NanoLC-MS/MS analysis of TMT labeled samples was performed on an Orbitrap Fusion Tribrid mass spectrometer (Thermo Fisher Scientific), coupled with a Dionex Ultimate 3000 RS nanoLC system (Thermo Fisher Scientific). Peptides were loaded on the trap column (75 $\mu$m × 2 cm PepMap-C18), using 0.1% TFA for 8 min with 10 $\mu$l/min flow rate. The peptides were then eluted and separated by an EASY-Spray column (75 $\mu$m × 50 cm RP-C18), at a constant flow rate of 300 nl/min. The gradient was as follows: 0–8 min, 1% B; 8–15 min, 1–10% B; 15–155 min, 10–32% B; 155–165 min, 32–75% B; 165–175 min, 75–95% B; 175–180 min, 95% B; buffer A (0.1% formic in water [vol/vol]) and buffer B (80% acetonitrile and 0.1% formic acid in water [vol/vol]). The MS data were acquired by Xcalibur control software (v 4.1.31.9) and the TMT-synchronous precursor selection-

MS3 method in top speed mode with cycle time 3 s. A scan cycle comprised MS1 scan from m/z range 375–1,600, with a maximum ion injection time of 50 ms and AGC value of 4 × 10$^5$, at a resolution of 120,000. The most intense ions were selected for fragmentation as MS2 using CID in the ion trap with 35% CID collision energy and an isolation window of 1.2 Th. The AGC target was set to 1.0 × 10$^4$ with a maximum injection time of 50 ms and a dynamic exclusion of 60 s, the scan rate was set to "Turbo." For accurate quantification of TMT peptides, a subsequent synchronous precursor selection-MS3 scan was performed. Five MS2 fragment ions were selected using with a window of 2 Th and further fragmented using HCD collision energy of 65%. The MS3 fragments were then analyzed in the Orbitrap with a resolution of 50,000, within m/z range 100–500. The AGC target was set to 5.0 × 10$^4$, and the maximum injection time was set to 120 ms. All MS data of TMT fractions were analyzed together by MaxQuant (v1.6.10.43) and searched against Homo sapiens database from UniProt (January 2020; SwissProt). The data were searched with the following parameters: fixed modification of carbamidomethyl (C), variable modifications of oxidation (M), and acetylation (protein N terminus), with maximum of two missed tryptic cleavages, reporter mass tolerance set to 0.03 ppm. The FDR threshold was set to 1% at peptide-spectrum match, peptides, and protein levels. The TMT quantification was set to reporter ion MS3 type with 10plex TMT (LOT: UH285228). The proteinGroups output table from MaxQuant was filtered in Perseus to remove "Potential contaminant," "Only identified by site," and "Reverse." Proteins with less than two peptides and without unique peptides were also removed.

## Phosphoproteomics

### Sample preparation

All samples were lysed in 2x LSB buffer (4% SDS, 14% glycerol, 200 mM DTT in 100 mM Tris–HCl, pH 6.8) to extract proteins, with 1x Pierce Protease Inhibitor Mini Tablets (EDTA-free, A32961; Thermo Fisher Scientific) and 1x phosphatase inhibitor (PhosSTOP, 4906845001; Roche) added. 200 $\mu$l lysis buffer extracted proteins. After being sonicated for 10 cycles (30 s on/off) using the Bioruptor Pico, the protein samples were centrifuged at 15,000$g$ for 15 min. The supernatant proteins were denatured at 95°C for 10 min, and then alkylated with 400 mM IAA (final concentration) in dark at room temperature for 40 min. The protein concentration was measured using EZQ Protein Quantitation Kit by following the manual. Proteins from each sample were further processed using SP3 protocol as described in Hughes et al (2019). In brief, protein samples were mixed with SP3 beads (1: 10; protein:beads) and digested with lysC/trypsin mixture (1: 50; enzyme:protein, A41007; Thermo Fisher Scientific). 200 $\mu$l 2% DMSO was used to elute the peptides from SP3 beads. The peptide concentration was measured using Pierce Quantitative Fluorometric Peptide Assay by following the manual. The peptide samples were dried and resuspended in the loading buffer for phosphopeptide enrichment. The phosphopeptides were enriched using the Fe-NTA Phosphopeptide Enrichment Kit (A32992; Thermo Fisher Scientific) by following the manual. 20 $\mu$l elution buffer was used to elute the phosphopeptides from the column. Dry the eluate immediately in a speed vacuum concentrator. The peptide samples were then re-dissolved in 5% FA and ready for LC–MS/MS analysis.

### LC–MS/MS analysis

All phosphopeptides samples were analyzed by using an Orbitrap Fusion Tribrid mass spectrometer (Thermo Fisher Scientific), equipped with a Dionex ultra-high-pressure liquid-chromatography system (RSLCnano). RPLC was performed using a Dionex RSLCnano HPLC (Thermo Fisher Scientific). Peptides were injected onto a 75 $\mu$m × 2 cm PepMap-C18 pre-column and resolved on a 75 $\mu$m × 50 cm RP-C18 EASY-Spray temperature-controlled (50°C) integrated column-emitter (Thermo Fisher Scientific), using a 3-h multistep gradient from 5% B to 35% B with a constant flow rate of 300 nl/min. The mobile phases were: $H_2O$ incorporating 0.1% FA (solvent A) and 80% ACN incorporating 0.1% FA (solvent B). The MS data were acquired under the control of Xcalibur software in a data-dependent acquisition mode using top speed and 3 s duration per cycle. The survey scan was acquired in the Orbitrap covering the m/z range from 375 to 1,600 kD with a mass resolution of 120,000 and an AGC target of $4.0 × 10^5$ ions. The most intense ions were selected for fragmentation using HCD 30% collision energy and an isolation window of 1.6 kD. The AGC target was set to $5.0 × 10^4$ with a maximum injection time of 54 ms and a dynamic exclusion of 60 s. The MS2 scan was acquired in the Orbitrap with a mass resolution of 30,000.

### MS data analysis

The data from all samples were analyzed together using MaxQuant (Tyanova et al, 2016b) v. 1.6.10.43, searched against the Homo sapiens database from UniProt (February 2023; SwissProt). The FDR threshold was set to 1% for each of the respective peptide-spectrum match and protein levels. The data were searched with the following parameters: stable modification of carbamidomethyl (C), variable modifications oxidation (M), acetylation (protein N terminus), and phosphorylation (STY), with maximum of 2 missed tryptic cleavages threshold. Lable-free quantification was selected.

### Western blotting and antibodies

The following antibodies were used: FAM111A (HPA040176, 1:500–1:1,000; Sigma-Aldrich, ab184572, 1:500–1:1,000; Abcam), PCNA (ab29, clone PC10, 1:1,000; ab18197, 1:1,000 for immunofluorescence; Abcam), H4K12ac (07-595, 1/1,000; Millipore), histone H3 (ab10799, 1:1,000; Abcam), MCM3 (ab4460, 1:1,000; Abcam), MCM2 (610701; 1:1,000; BD Bioscience, ab4461 1:1,000 for immunofluorescence; Abcam), CDC45 (CST #11881, 1:1,000), RPA2 (ab2175; clone 9H8, 1:1,000, 1:300 for immunofluorescence), histone H4 (05-858, 1:1,000; Millipore), GINS1 (1:500; kind gift from Karim Labib's lab), p-CHK1 S345 (CST #2348; 1:1,000), CHK1 (sc-56291, 1:1,000; Santa Cruz), p-CHK2 T68 (CST #2661, 1:1,000), p-RPA S4/S8 (A300-245A-M 1:1,000; Bethyl), p-H2A.X S139 (γH2A.X) (CST #2577, 1:1,000, 1:500 for immunofluorescence), p-RPA S33 (A300-246A, 1:1,000; Bethyl), anti-HA (CST #3724, 1:1,000; BioLegend 901501, 1:000 for immunofluorescence), GAPDH (CST #2118, 1:1,000), CDC45 (CST #11881, 1/50), FAM111B (PA5-58474, 1:1,000–1:2,000; Invitrogen). Secondary antibodies conjugated with HRP were from Jackson ImmunoResearch Labs. Fluorescent dye–conjugated antibodies were sourced from Li-Cor. Signals were revealed by chemiluminescence substrate from Pierce (SuperSignal West Pico or SuperSignal West Femto) and imaged using ChemiDoc XRS+ or Licor Odyssey imaging systems. Uncropped gel scans for Figs 1A and D, 4B, and S2C are provided in Fig S5.

### Computational protein sequence analysis for identification of FAM111A UBL repeats

Multiple sequence alignments were generated with T-Coffee using default parameters (Notredame et al, 2000), slightly refined manually and visualized with the Belvu program (Sonnhammer & Hollich, 2005). Sequences were named according to their UniProt identifiers (Wu et al, 2006). Profiles of the alignment as global hidden Markov models (HMMs) were generated using HMMer (Eddy, 1996; Finn et al, 2011). Profile-based sequence searches were performed against the Uniref50 protein sequence database (Wu et al, 2006) using HMMsearch (Eddy, 1996; Finn et al, 2011). Remote homology analyses were performed using HHpred profile-to-profile comparisons (Soding et al, 2005). Profile-to-profile (HHpred) matches were evaluated in terms of an E-value, which is the expected number of non-homologous proteins with a score higher than that obtained for the database match. An E-value much lower than one indicates statistical significance. Secondary structure predictions were performed using PsiPred (Jones, 1999). The final UBL alignment was obtained using a combination of profile-to-profile comparisons (Soding et al, 2005) and sequence alignments derived from structural superpositions of a selection of UBL domains whose tertiary structure is known (PDB IDs: 1WFY, 5J2R, 2D07, and 3EEC) (Holm & Sander, 1995). Figures were generated using Inkscape (http://inkscape.org/). Structures and 3D models were analyzed using PyMol (http://www.pymol.org). Structural models were created using MODELLER (Sali & Blundell, 1993). Surface electrostatic potential representations were generated using PyMol APBS (Adaptive Poisson-Boltzmann Solver) interface and were colored according to charge levels ranging from –5 kT/e (red) to +5 kT/e (blue).

### Statistical analysis

For statistical analysis, unpaired $t$ tests and Mann–Whitney tests were performed using Prism.9. $P$-values are indicated by asterisks ($P < 0.0001$ [****], $P < 0.001$ [***], $P < 0.01$ [**], and $P < 0.05$ [*]), and n.s. indicates non-significant.

# Data Availability

Mass spectrometry proteomic raw data are available at the proteomeXchange Consortium database via the proteome identifications (PRIDE) repository under the dataset ID PXD044180.

# Supplementary Information

# Acknowledgements

We thank Julian Blow, Anja Groth, Karim Labib, Chris Ponting, John Rouse, Giulia Saredi, Jordan Taylor, and Jane Wright for comments on the manuscript. We thank Anja Groth for the GFP-RPA1 and RFP-PCNA U-2-OS cell line,

Karim Labib for the GINS1 antibody, John Rouse for the Flp-In T-Rex U-2-OS cell line, and the Tobias Meyer laboratory for the CDC45 immunofluorescence conditions. We thank the Centre for Advanced Scientific Technologies (CAST) for access to the FACS facility, the Proteomics and Mass Spectrometry Facility FingerPrints, the Dundee imaging facility, the Drug Discovery Unit for access to the Operetta high content microscope, and the MRC PPU reagents and services facility. We would like to thank Manu Decker, Nina Svensen, Giulia Saredi, and Federico Tinarelli for technical assistance. DO Rios-Szwed acknowledges support from MRC PhD studentship, V Alvarez and E Garcia-Wilson received support from CRUK, S Bandau received support from ERC. Research in Alabert laboratory is supported by CRUK Career Development fellowship C57404/A21782 and European Research Council ERC-STG715127. Research in Lamond laboratory is supported by Wellcome Trust Collaborative Award 206293/Z/17/Z. Research in Ponting laboratory is supported by the MRC MC_UU_00007/15.

## Author Contributions

DO Rios-Szwed: conceptualization, data curation, formal analysis, investigation, methodology, and writing—original draft, review, and editing.

V Alvarez: data curation, formal analysis, investigation, methodology, and writing—original draft.

L Sanchez-Pulido, H Jiang, and S Bandau: formal analysis and investigation.

E Garcia-Wilson: data curation, formal analysis, and investigation.

A Lamond: formal analysis, funding acquisition, and writing—review and editing.

C Alabert: conceptualization, resources, data curation, formal analysis, supervision, funding acquisition, validation, investigation, visualization, methodology, and writing—original draft, review, and editing.

## Conflict of Interest Statement

The authors declare that they have no conflict of interest.

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
