## [Reviewer comments · Life Science Alliance]

Life Science Alliance

FAM111A regulates replication origin activation and cell fitness

Diana Rios Szwed, Vanesa Alvarez, Luis Sanchez Pulido, Elisa Garcia Wilson, Hao Jiang, Susanne Bandau, Angus Lamond, and Constance Alabert

DOI: <https://doi.org/10.26508/lsa.202302111>

Corresponding author(s): Constance Alabert, University of Dundee and Diana Rios Szwed, University of Edinburgh

Review Timeline:

Submission Date:	2023-04-25
Editorial Decision:	2023-04-25
Revision Received:	2023-08-01
Editorial Decision:	2023-08-29
Revision Received:	2023-09-11
Editorial Decision:	2023-09-11
Revision Received:	2023-09-19
Accepted:	2023-09-19

Transaction Report:

Please note that the manuscript was reviewed at *Review Commons* and these reports were taken into account in the decision-making process at *Life Science Alliance*.

Review #1

Rios-Szwed and colleagues investigate functions of FAM111A, a protease that Dr. Alabert has previously shown to localize at nascent DNA and promote PCNA loading. In this manuscript, the authors first describe that FAM111A facilitates efficient activation of replication origins by using DNA combing experiments and by analyzing chromatin loading of DNA replication proteins. Next, they show that FAM111A KO cells show reduced levels of ssDNA exposure after replication stress. Then the authors move on to show that the major FAM111A interactor is FAM111B, which they show to localize at nascent DNA and is epistatic to FAM111A in promoting DNA replication as well as RPA loading after replication stress. Finally, the authors show that unregulated FAM111A activity, either by overexpression of WT FAM111A or disease-associated mutants, causes extensive exposure of ssDNA.

****Major comments****

1. Fig. S1G: Actual inter-origin distances (distance between replication tracks in which a CldU track is flanked by IdU tracks on both sides) should be plotted to estimate the changes in origin firing frequencies. The results should be presented as inter-origin distances, not ratios between UCN-01-treated and untreated. The revised experiment should be included in the main figures as this is central to the conclusion, and statistics should be included.
2. The claim "FAM111A ... promotes DNA replication initiation of active and dormant origins" (page 4, line 4) is not fully supported by experiments. Does FAM111A localize at replication origins? Without direct evidence of FAM111A being present at replication origins, it remains possible that the changes in origin activity is secondary to the loss of FAM111A function at forks or something else.
3. Fig. S1G: If FAM111A's function to promote activation of dormant origins in response to UCN-01 is unrelated to the function of FAM111A at forks, it is expected to be independent of the PIP motif. Is it the case?
4. Fig. 2B: Increased survival after HU treatment might be secondary to reduced S-phase populations in FAM111A-depleted cells (Fig. 1C) as HU would affect only S-phase cells.
5. Fig. 2B-I: Similarly, the blunted response to replication stress in FAM111A depleted cells could be simply explained by reduced number of forks per cell as indicated by increased inter-fork distance (Fig. 1F). Similarly, the authors' group has previously reported reduced PCNA levels on chromatin (Alabert et al, 2014), suggesting that there are reduced number of active forks per nucleus.
6. Fig. 2H "FAM111A depletion reduced ssDNA exposure upon HU treatment (Fig. 2H, 2I)": The figure in Fig. 2H does not appear to be treated with FAM111A RNAi. If this is FAM111A RNAi cells, siControl cells need to be shown as a comparison.
7. Fig. 3B,C: The interaction between FAM111A and FAM111B needs to be validated by coimmunoprecipitation-WB of endogenous proteins.
8. Fig. 4A-C: Induction of DNA damage and apoptosis by FAM111A WT and disease mutants (including T338A that the authors claim unstudied) has been reported by Hoffman et al. and therefore not novel.
9. Fig. 4E: The increase in ssDNA intensities is mild and might not be biologically significant.
10. Fig. 4G: Cell cycle status needs to be assessed by FACS after treatment with each drug. Bleomycin might induce G1/S arrest if G1/S checkpoint is intact.
11. ssDNA exposure after FAM111A OE might not be because FAM111A has a function in promoting ssDNA exposure, but could be simply explained by replication fork stalling, for example, due to degradation of essential proteins as proposed before (Hoffman et al, 2020).
12. Page 8, line 17, "Altogether, these data revealed that unrestrained FAM111A peptidase activity leads to ssDNA exposure upstream of apoptosis.": Just because the caspase inhibitor did not block the ssDNA exposure, it does not mean ssDNA exposure is upstream of apoptosis - it could be happening in parallel and might be unrelated. A similar unsupported conclusion "ssDNA exposure is upstream of apoptosis" appears in other places: page 8, line 30; page 9, line 22.
13. Whether protease activity is necessary for the FAM111A function in regulation of origin activation and in ssDNA exposure is not addressed. Can the phenotypes of FAM111A KO cells be rescued by FAM111A WT but not an active site mutant?
14. Similarly, the authors need to test whether the PIP motif of FAM111A is required for the function of FAM111A at forks, such as promoting ssDNA exposure.

****Minor comments****

1. Page 2, Line 8, "FAM111A catalytic activity has not been shown in vitro": Protease activity of FAM111A has been shown using recombinant proteins in vitro by Hoffman et al, 2020.
2. Page 7, line 26, "T338A is a previously unstudied GCLEB patient mutation.": The T338A mutant was studied by Hoffman et al. and shown to have hyperactivity in vitro and to cause DNA damage when overexpressed in cells.

****Referee cross-commenting****

I feel that this study has problems even as a descriptive study. As I mentioned in my review, there are alternative explanations for their observations that the authors have not ruled out. If the authors remove all unsupported claims, then there is not much to conclude from this study. I am not saying their conclusions are wrong - I think this study is just premature.

This study could be of interest to the audience in DNA replication/DNA repair field and could be unveiling a new function of FAM111A in DNA replication. However, in the current form, this study appears to be a collection of loosely connected observations of FAM111A-manipulated cells without a clear message of what FAM111A does at replication forks and origins. Each observation appears to be loosely tied together with a keyword of ssDNA exposure, but how FAM111A regulates or changes ssDNA exposure is not addressed. The described phenotypes are potentially interesting, but for each observation there is an alternative explanation that could affect authors' interpretation. As outlined in my comments, lack of mechanism, lack of clear conclusion, and misinterpretation of some of the data led to this less enthusiastic review.

Review # 2

The manuscript by Rios-Szwed et al have investigated the role of FAM111A in DNA replication. Previous studies had identified that FAM111A suppresses DNA replication via an interaction with RFC and that hyperactive mutants induce apoptosis. Now, Rios-Szwed et al discovered that FAM111A knockdown affects inter origin distance without checkpoint induction. In particular, the firing of dormant origins when dNTPs are limiting is suppressed and less ssDNA is produced. Although FAM111B is a strong interactor of FAM111A, no additive effect on DNA replication was detected when both proteins were depleted. On the other hand, overexpression or hyperactive mutants promote more gammaH2AX and ssDNA even in the presence of a caspase inhibitor, suggesting that the protease functions in ssDNA production prior to apoptosis.

****Major comments:****

Dormant origins are frequently inhibited by phosphatases - is there any evidence that phosphatases are the target of FAM111A. In this context I would suggest to blot for Treslin, as it is one of the first factors being recruited in a kinase dependent manner to the MCM2-7 complex.

****Minor comments:****

Abstract: Unclear why too much FAM111A causes cell death

Introduction: the R569H point mutant needs to be better introduced - e.g. explain where the mutation is localised or what it affects e.g. it is localised in the predicted peptidase domain

Figure 1A and 1D - are all the lanes shown originating from the same gel - if not please repeat.

Page 3 - I am not sure that in FAM111A depleted cells the DNA synthesis rate is reduced. Could it be, that just fewer cells are in S-phase.

Page 3 - It is stated: "In contrast, the inter-fork distance was slightly increased in FAM111A depleted cells (Fig. S1E)", however, the data but the data do not fully support this statement.

Figure 4C - the quantification of the last lane looks wrong. Is the average or the median? Please find information in the figure and methods section.

Question: If both FAM111A and FAM111B are overexpressed - is this better tolerated?

Is there a homologue in other species?

****Referee cross-commenting****

I agree with the other reviewers that the study has a descriptive nature. I guess this could be acceptable dependent on the journal choice.

In general, I really like the study as it establishes how initiation of DNA replication is affected by inhibition and activation of FAM111A. The work is done well and deserves to be seen in a good journal.

The study helps the field to move forward and will allow a more targeted search for specific protease targets. In this way it will help clinicians and also researchers.

My expertise is in initiation of DNA replication.

Review #3

Rios-Szwed and co-authors show that the depletion of FAM111A results in faster replication speed, longer intra-origin distances, and less chromatin-bound RPA even without induction of replication stress in U2OS cells. Induction of replication stress in FAM111A-depleted cells results in blunted response with less DNA damage, decreased checkpoint activation and resistance to the replication-stress inducing agent, HU. They show that cells without FAM111A display lower levels of single stranded DNA after treatment.

In the second part, the authors show that FAM111A and FAM111B form a complex, although the similarities and differences of their functions are not explored in detail. From the little data shown, it looks like they might be working together in controlling amount of ssDNA. They find that both proteins are expected to have two conserved UBL domains, with one of them overlapping with ssDNA binding domain. Finally, the authors use overexpression of WT and mutant proteins to show that expression of WT and patient-derived mutant has increased level of DNA damage, increased levels of ssDNA, with and without DNA damage, and that the peptidase domain is necessary for the phenotypes.

The data from the first two figures are consistent with FAM111A being involved in regulation of single stranded DNA formation during normal replication and during replication stress. Unfortunately, the work gives no indication of the mechanism of such regulation. I am not convinced that the function has much to do with controlling origin activation (see below). The data from the last two figures is also descriptive. Until the substrates of FAM111A are identified, there will be no understanding of its true function and the data will continue to be descriptive.

****Specific points:****

Figure 1:

The siFAM111A-2 has a stronger phenotype in growth assay but has very little change in levels of cells in G1. No complementation of the phenotype is given.

1D- there is no total RPA so it is unclear if there is no change in pRPA in relation to total RPA. Small differences will be missed without DNA damage and it would be helpful to use more sensitive assays to identify the reduction in ssDNA under unperturbed conditions.

1E- what does the data look like if the lengths of ldu are plotted? This would be a measure of speed of the ongoing forks. Generally, this would be better than the CldU measurement.

1F- the Inter-CldU distance increase could be secondary (indirect effect) of the increased replication speed

1G- It looks like there are many more data points in the siFAM111A-1 and many fewer in the siFAM111A-2. The increase in the MCM quantified in H is bigger with si2 even though the G1 distribution has less change than with si1. Consequently, these data are inconclusive.

1I- no plot is shown for si2 but it is quantified. It would be informative to see the plots for easy comparison.

Figure 2:

This is the most interesting part of the paper and generally is well done. As mentioned above, I believe that the phenotype the authors see in Figure 1 is the same phenotype as seen here- less production of ssDNA but it is hard to see this under unperturbed conditions, thus more data should be gathered to test that.

Figure 3: shows novel findings but it is unclear how it relates to the rest of the paper except that it suggests that the paralogs may work together in the pathway that has been explored in Figure 1 and 2. The authors perform computational and predictive analysis that identifies two UBL domains in the FAM111A/B paralogs. The FAM111A UBL2 domain is known to bind ssDNA. The authors might test if the domain can also bind ssDNA in FAM111B and if FAM111B has similar ability to promote ssDNA formation

Figure 4:

The human mutations provide some insight as to the requirement for functional peptidase activity for the function of the protein. The work would also be strengthened if a ssDNA binding mutant was made and tested given the authors interest in defining the UBL domains.

Not sure why they use a term "ssDNA exposure"? It implies a removal of something that was covering it which they

certainly do not show. I would use ssDNA levels, maybe ssDNA production, formation?

****Other points:****

As QIBC is used throughout the paper, it would be nice to have a brief explanation of the technique when it is first introduced.

The authors write that the function of FAM111A in promoting ssDNA formation is "distinct from overcoming protein-DNA complexes ahead of the replisome by Top1 or PARP1". It is not clear to this reader how they have determined that they are not the result of the same mechanism as the phenotypes seem very related. I would clarify this point.

Since the authors are including patient mutations, more introduction to the diseases would be useful.

****Referee cross-commenting****

I have no further comments.

The findings add to the growing literature on the FAM111 proteins and will be of interest to scientists who are studying them and those interested in replication and replication stress response.

April 25, 2023

Re: Life Science Alliance manuscript #LSA-2023-02111-T

Dr. Constance Alabert
MCDB, University of Dundee
School of Life Sciences
Dow street
Dundee, Scotland DD15EH
United Kingdom

Dear Dr. Alabert,

Thank you for submitting your manuscript entitled "FAM111A regulates replication origin activation and cell fitness" to Life Science Alliance. We invite you to re-submit the manuscript, revised according to your Revision Plan.

Thank you for this interesting contribution to Life Science Alliance. We are looking forward to receiving your revised manuscript.

Sincerely,

B. MANUSCRIPT ORGANIZATION AND FORMATTING:

1. NEW EXPERIMENTS ADDED TO THE REVISED MANUSCRIPT.

Reviewer 1

R1.4. Fig. 2B: Increased survival after HU treatment might be secondary to reduced S-phase populations in FAM111A-depleted cells (Fig. 1C) as HU would affect only S-phase cells. **R1.5.** Fig. 2B-I: Similarly, the blunted response to replication stress in FAM111A depleted cells could be simply explained by reduced number of forks per cell as indicated by increased inter-fork distance (Fig. 1F). Similarly, the authors' group has previously reported reduced PCNA levels on chromatin (Alabert et al, 2014), suggesting that there are reduced number of active forks per nucleus. We have rewritten the result section to highlight that the origin activation defect observed in unperturbed condition and upon UCN-01 treatment (Figure 1) could explain the resistance to HU treatment (Figure 2). Moreover, we have measured ssDNA formation upon UCN-01 treatment in control cells and upon FAM111A depletion (New Figure S2E), to further highlight the probable similarity between UCN-01 and HU treatments. Regarding the difficulty of detecting changes in unperturbed conditions, using QIBC we have shown that ssDNA formation is reduced upon FAM111A depletion (Fig. 2I).

R1.7. Fig. 3B,C: The interaction between FAM111A and FAM111B needs to be validated by coimmunoprecipitation-WB of endogenous proteins. This experiment is now included in the revised version of the manuscript as new Figure S3C.

R1.10. Fig. 4G: Cell cycle status needs to be assessed by FACS after treatment with each drug. Bleomycin might induce G1/S arrest if G1/S checkpoint is intact. These control experiments have been performed by QIBC and included in the revised manuscript as new Figure 4I.

R1.14. Similarly, the authors need to test whether the PIP motif of FAM111A is required for the function of FAM111A at forks, such as promoting ssDNA exposure. ssDNA formation upon overexpression of FAM111A PIP mutant is shown in new Figure 4H and Figure S5H. It revealed that FAM111A binding to PCNA is important but not essential to promote ssDNA formation. It complements the finding that S phase entry and the protease activity of FAM111A are necessary to promote ssDNA formation (Fig. 4F, I-J).

Reviewer 2

R2.1 Dormant origins are frequently inhibited by phosphatases - is there any evidence that phosphatases are the target of FAM111A. In this context I would suggest to blot for Treslin, as it is one of the first factors being recruited in a kinase dependent manner to the MCM2-7 complex. The abundance of 152 phosphatases has been examined upon FAM111A OE by mass spectrometry and none of them were significantly affected by FAM111A OE (Table S2). Based on Reviewer 2 suggestion, we have performed a phosphoproteomic analysis of the level of phosphorylation of proteins upon FAM111A depletion, and a handful of proteins linked to DNA replication, such as RIF1 and ORCA, were hyperphosphorylated upon FAM111A depletion

(New Figure S4C, new Table S3). Of note, none of the phosphorylation sites identified have known roles in the initiation of DNA replication.

R2.4 Figure 1A and 1D - are all the lanes shown originating from the same gel - if not please repeat. All the lanes are from the same gel. Full scans are included in the revised version of the manuscript as new Figure S6.

R2.8 Question: If both FAM111A and FAM111B are overexpressed - is this better tolerated? The simultaneous OE of FAM111A and FAM111B mirrors the effect of FAM111A OE alone (New Figure S4E, lane 3 and 4). It suggests that the toxicity of FAM111A OE is not due to changes in the ratio of the two proteins in these conditions.

Reviewer 3

R3.2 1D- there is no total RPA so it is unclear if there is no change in pRPA in relation to total RPA. The total RPA control is included in the revised version of the manuscript (New Figure S1D). We have also shown by QIBC that total RPA levels are not reduced upon FAM111A depletion but rather increased (Figure S1M). Small differences will be missed without DNA damage and it would be helpful to use more sensitive assays to identify the reduction in ssDNA under unperturbed conditions. In unperturbed conditions, QIBC has been used to measure ssDNA level (Fig. 2I, minus HU), and RPA level on chromatin in S phase cells (Fig. S1L). Both analyses revealed that FAM111A depletion reduces ssDNA formation and RPA abundance on chromatin.

R3.3 1E- what does the data look like if the lengths of IdU are plotted? This would be a measure of speed of the ongoing forks. Generally, this would be better than the CldU measurement. IdU track lengths were also measured and showed the same result as for CldU measurement. These data are now included in the revised manuscript as New Figure S1E.

R3.6 1I- no plot is shown for si2 but it is quantified. It would be informative to see the plots for easy comparison. CDC45 levels in siFAM111A-2 have been added in the revised version of the manuscript as new Figure 1I.

R3.7. This is the most interesting part of the paper and generally is well done. As mentioned above, I believe that the phenotype the authors see in Figure 1 is the same phenotype as seen here- less production of ssDNA but it is hard to see this under unperturbed conditions, thus more data should be gathered to test that. Please see Reviewer 1.4 above and new Figure S2E in the revised version of the manuscript.

R3.8. Figure 3: shows novel findings but it is unclear how it relates to the rest of the paper except that it suggests that the paralogs may work together in the pathway that has been explored in Figure 1 and 2. The authors perform computational and predictive analysis that identifies two

UBL domains in the FAM111A/B paralogs. The FAM111A UBL2 domain is known to bind ssDNA. The authors might test if the domain can also bind ssDNA in FAM111B and if FAM111B has similar ability to promote ssDNA formation. We do not have the expertise to test if the UBL2 domain in FAM111B can also bind ssDNA. However, we have tested whether FAM111B has similar ability to promote ssDNA formation (New Figure 4G). Like γ H2AX (Fig. 3K), FAM111B expression does not increase ssDNA formation.

2. NEW SECTIONS ADDED TO THE REVISED MANUSCRIPT AND CLARIFICATIONS.

Reviewer 1

R1.1. Fig. S1G: Actual inter-origin distances (distance between replication tracks in which a CldU track is flanked by IdU tracks on both sides) should be plotted to estimate the changes in origin firing frequencies. The results should be presented as inter-origin distances, not ratios between UCN-01-treated and untreated. The revised experiment should be included in the main figures as this is central to the conclusion, and statistics should be included. Measures of inter fork distance are provided instead of inter origin distance (IOD) because not enough values for IOD were detected per coverslips (~10 IOD per coverslips, >200 inter fork distances). As IOD values are often widely scattered, a large number of measures (at least 100), are required for a reliable estimation of IOD (Techer et al. 2013). Regarding the ratios, inter fork distance measures for untreated and UCN-01 treated cells are provided in Figure 1F and Figure S1E. Ratios are used to estimate the effect of FAM111A depletion on dormant origins independently of the effect observed on origins in unperturbed conditions.

R1.2. The claim "FAM111A ... promotes DNA replication initiation of active and dormant origins" (page 4, line 4) is not fully supported by experiments. Does FAM111A localize at replication origins? Without direct evidence of FAM111A being present at replication origins, it remains possible that the changes in origin activity is secondary to the loss of FAM111A function at forks or something else. Using the endogenous FAM111A IP conditions from chromatin fractions (Fig. 3C), we have performed FAM111A Chip-seq experiments but did not obtain results on time for this revision. Therefore, it remains unknown whether FAM111A directly or indirectly promote origin activation. This point is clearly stated in the discussion section of the revised version of the manuscript, p10: *Our data revealed that FAM111A promotes origin activation and ssDNA formation. Both functions could be achieved indirectly by degrading a protein preventing S phase entry, as both mechanisms rely on S phase entry.*

R1.6. Fig. 2H "FAM111A depletion reduced ssDNA exposure upon HU treatment (Fig. 2H, 2I)": The figure in Fig. 2H does not appear to be treated with FAM111A RNAi. If this is FAM111A RNAi cells, siControl cells need to be shown as a comparison. Figure 2H is siControl +/- HU to present the setup of the ssDNA assay. Figure 2I is the quantification for siControl, siFAM111A

in +/- HU. We have clarified this point in the result section (p5): *“To do so we used BrdU labeling and detection in non-denaturing conditions (Mejlvang et al., 2014) (Fig. 2H, S2D). Like the phenotypes observed in RPA experiments, FAM111A depletion reduced ssDNA exposure upon HU treatment (Fig. 2I).”*

R1.8. Fig. 4A-C: Induction of DNA damage and apoptosis by FAM111A WT and disease mutants (including T338A that the authors claim unstudied) has been reported by Hoffman et al. and therefore not novel. **R1.16.** Page 7, line 26, "T338A is a previously unstudied GCLEB patient mutation.": The T338A mutant was studied by Hoffman et al. and shown to have hyperactivity in vitro and to cause DNA damage when overexpressed in cells. We have amended the revised manuscript, p8: *“...the S541A mutation generates a FAM111A putative peptidase dead mutant and the R569H, Y511H and T338A mutations potentiate FAM111A peptidase activity (Hoffmann et al., 2020; Kojima et al., 2020).”*

R1.11. ssDNA exposure after FAM111A OE might not be because FAM111A has a function in promoting ssDNA exposure, but could be simply explained by replication fork stalling, for example, due to degradation of essential proteins as proposed before (Hoffman et al, 2020). This possibility has been added in the discussion together with Hoffman et al, 2020 reference, p10: *Alternatively, FAM111A may directly degrade an essential protein, leading to fork stalling (Hoffmann et al., 2020) or degrade a DNA binding protein(s) blocking origin activation and ssDNA exposure. The latter is reminiscent of...”*

R1.12. Page 8, line 17, "Altogether, these data revealed that unrestrained FAM111A peptidase activity leads to ssDNA exposure upstream of apoptosis.": Just because the caspase inhibitor did not block the ssDNA exposure, it does not mean ssDNA exposure is upstream of apoptosis - it could be happening in parallel and might be unrelated. A similar unsupported conclusion "ssDNA exposure is upstream of apoptosis" appears in other places: page 8, line 30; page 9, line 22. Introduction, result, and discussion sections have been rewritten to clarify that *“ssDNA is not caused by apoptosis”* instead of *“ssDNA exposure is upstream of apoptosis”*.

R1.15. Page 2, Line 8, "FAM111A catalytic activity has not been shown in vitro": Protease activity of FAM111A has been shown using recombinant proteins in vitro by Hoffman et al, 2020. The manuscript has been amended, p2: *“FAM111A catalytic activity has been shown in vitro and recent work revealed that in cellulo FAM111A exhibits autocleavage activity when its peptidase domain is intact (Hoffmann et al., 2020; Kojima et al., 2020).”*

Reviewer 2

R2.2 Abstract: Unclear why too much FAM111A causes cell death. *This point is discussed in the introduction (p2), and the discussion sections (p10).*

R2.3 Introduction: the R569H point mutant needs to be better introduced - e.g. explain where the

mutation is localised or what it affects e.g. it is localised in the predicted peptidase domain. **R3.13.** Since the authors are including patient mutations, more introduction to the diseases would be useful. We have amended the introduction section as followed, p2: “In humans, heterozygous point mutations in FAM111A are linked to two severe developmental syndromes: the Kenny-Caffey syndrome (KCS2, OMIM-127000) and Gracile Bone Dysplasia (GCLEB, OMIM-602361). In both diseases, patients are characterized by, among others, short stature, hypocalcemia, hypoparathyroidism and dense or gracile bones (Welter and Machida, 2022). Heterozygous de novo mutations are the most common, and AlphaFold predicted structure of FAM111A reveals that patient mutations are located within two clusters, within the enzyme domain, and in a flexible region between the ssDNA binding domain and the enzyme domain. Remarkably, the R569H point mutation is located outside of the enzyme domain, in the C terminal region of the FAM111A gene and is found in seven unrelated KCS2 patients,...”

R2.5 Page 3 - I am not sure that in FAM111A depleted cells the DNA synthesis rate is reduced. Could it be, that just fewer cells are in S-phase. In Fig. S1A, EdU intensities are measured in EdU positive cells. Therefore, the proportion of S phase cells does not affect the measure of DNA synthesis rate. This point has been clarified in the labelling of the Y axis and the legend of Figure S1A in the revised manuscript.

R2.6 Page 3 - It is stated: "In contrast, the inter-fork distance was slightly increased in FAM111A depleted cells (Fig. S1E)", however, the data but the data do not fully support this statement. We have amended the manuscript accordingly (p3):“In contrast, the inter-fork distance was increased in FAM111A depleted cells, although not significantly in all conditions (Fig. S1G). Therefore, to further test the possibility that fewer origins had initiated upon FAM111A depletion, we artificially triggered dormant origin activation with the CHK1 inhibitor 7-hydroxystaurosporine (UCN-01)...”

R2.7 Figure 4C - the quantification of the last lane looks wrong. Is the average or the median? Please find information in the figure and methods section. The red bar is the mean, and the statistical test used in Unpaired student T test. This information has been updated in figure legend and method section. Figure legend, p14, C, E-H and J, red bars, mean; Unpaired t test, ****P < 0.0001, *P < 0.05, n.s. non-significant. Method section, p23: P-values are indicated by asterisks (P < 0.0001 (****), P < 0.001 [***], P < 0.01 [**], and P < 0.05 [*]), and n.s. indicates non-significant.

R2.9 Is there a homologue in other species? We have generated a conservation tree using tools from the NGPhylogeny.fr server (PMID: 31028399). The origin of FAM111 seems to go back to invertebrates (cnidarians), but it has been lost in many branches, such as: nematodes, molluscs, arthropods, and echinoderms. FAM111 was duplicated in the origin of placental mammals (FAM111A and FAM111B), but the FAM111B subfamily was later lost in the glires clade, which includes rodents (mouse) and lagomorphs (rabbits). In teleosts (most fishes), FAM111

gene has undergone different duplication events in multiple species (this is common in teleosts). Because our analysis is not substantially different from the one recently published by Welter and Machida (Welter and Machida, 2022) we did not include it in the manuscript.

Reviewer 3

R3.1. The siFAM11A-2 has a stronger phenotype in growth assay but has very little change in levels of cells in G1. No complementation of the phenotype is given. siFAM11A-2 depletion is more efficient than siFAM11A-1 (Fig. 1A). siFAM11A-2 depletion leads to more severe DNA synthesis (Fig. S1A), CDC45 loading (Fig. 1J) and viability (Fig. S1B) defects. However, as pointed by Reviewer 3, one phenotype is less severe in siFAM11A-2 compared to siFAM11A-1, the proportion of cells in G1 phase (Fig. 1C). One possibility is that siFAM11A-1, but not siFAM11A-2, leads to DNA damage accumulation in G1 phase. Yet, neither siFAM11A-1 nor siFAM11A-2 led to increased level of DNA damage in G1 phase (panel B below). Another possibility is that siFAM11A-2 induces more severe DNA replication defects compared to siFAM11A-1, extending S phase length, and thereby disturbing the distribution of each cell cycle phase. To test this possibility, we have measured S phase length upon FAM11A depletion using the successive EdU and BrdU labelling strategy (Bialic et al, 2022). We found that S phase length is more severely affected by siFAM11A-2 compared to siFAM11A-1 (13h in siControl, 15h for siFAM11A-1 and 20h for siFAM11A-2, panel A below). The longer S phase is consistent with the stronger effect on CDC45, RPA and ssDNA levels. As cell cycle data in Figure 1C is presented as a proportion of total cells, it stands to reason that greater percentage of cells in S phase would lead to smaller increase in the fraction on cells in G1 phase.

A. S phase length measured according to the EdU/BrdU double labelling flow cytometry-based strategy described in Bialic et al. 2022. **B.** 53BP1 foci quantification in G1 phase cells. As a positive control, cells were treated with aphidicolin for 6 h (siCtl + APH). >400 nuclei were counted per condition.

R3.4 1F- the Inter-CldU distance increase could be secondary (indirect effect) of the increased replication speed. To separate cause and effect between origin activation and fork speed, we have

used UCN-01 treatments. UCN-01 promotes promiscuous origin activation, and therefore in these conditions, origins fire independently of fork speed. The fact that fork density remains low in FAM111A depleted cells compared to control cells is a good indication that the primary defect upon FAM111A depletion is an origin activation defect rather than fork speed.

R3.5 1G- It looks like there are many more data points in the siFAM11A-1 and many fewer in the siFAM11A-2. The increase in the MCM quantified in H is bigger with si2 even though the G1 distribution has less change than with si1. Consequently, these data are inconclusive. Using 3D gating strategies (EdU, DAPI and MCM2, Gardner et al. 2017), G1 phase cells can be identified as G1 DNA content, EdU negative. In Figure 1G, chromatin bound MCM2 level are measured in G1 phase cells, not in total cells. Therefore, the number of nuclei analyzed and the proportion of cells in G1 phase does not affect this measure.

R3.10. Not sure why they use a term "ssDNA exposure"? It implies a removal of something that was covering it which they certainly do not show. I would use ssDNA levels, maybe ssDNA production, formation? It is a term used in DNA replication stress studies, although ssDNA formation is used more frequently. We have modified the manuscript accordingly.

R3.11. As QIBC is used throughout the paper, it would be nice to have a brief explanation of the technique when it is first introduced. A brief explanation is included in the result section of the transferred manuscript (p4), together with the reference study from Jiri Lukas laboratory that used QIBC to measure RPA level, Toledo et al, Cell 2013. "QIBC provides measures of the intensity of a protein by immunofluorescence, at the single cell level and in thousands of cells, bridging the gap between microscopy and flow cytometry (Toledo et al., 2013)".

R3.12. The authors write that the function of FAM111A in promoting ssDNA formation is "distinct from overcoming protein-DNA complexes ahead of the replisome by Top1 or PARP1". It is not clear to this reader how they have determined that they are not the result of the same mechanism as the phenotypes seem very related. I would clarify this point. Top1 and PARP1 degradations mediated by FAM111A relies on replication fork arrested by a DNA Protein Crosslink (DPC), constituting a roadblock for replisome progression. Here we monitor FAM111A function in HU treated cells, where the replisome is arrested due to nucleotide production defects. This sentence emphasizes this possible difference. As pointed by reviewer 3, we cannot exclude that it can be a related mechanism. Therefore, we have amended this section as follow, p6: This novel function may be distinct from FAM111A role in overcoming protein-DNA complexes ahead of replisomes formed by Topoisomerase 1 (Kojima et al., 2020) or PARP1 (Murai et al., 2012; Murai et al., 2014), as in either HU or APH treated cells, replisomes are not arrested due to obstacles ahead of the fork.

3. DESCRIPTION OF ANALYSES NOT CARRIED OUT

Reviewer 1

R1.3. Fig. S1G: If FAM111A's function to promote activation of dormant origins in response to UCN-01 is unrelated to the function of FAM111A at forks, it is expected to be independent of the PIP motif. Is it the case? This possibility has not been tested as we do not have the adequate cell system to do so. We have a TET inducible PIP mutant version of FAM111A (Fig. 4), not a KI cell line. Therefore, this analysis could not be provided within the scope of this revision.

R1.13. Whether protease activity is necessary for the FAM111A function in regulation of origin activation and in ssDNA exposure is not addressed. Can the phenotypes of FAM111A KO cells be rescued by FAM111A WT but not an active site mutant? We show that the protease activity is necessary for ssDNA formation, as OE of FAM111A peptidase dead does not lead to increased ssDNA formation (Fig. 4). We do not have the tool to test if the protease activity is necessary for FAM111A function in origin activation. Several attempts to rescue FAM111A activity through ectopic expression were deleterious to cells, as we have shown in figure 4; Hence our inability to carry out rescue experiments in the current system.

Reviewer 3

R3.9 The human mutations provide some insight as to the requirement for functional peptidase activity for the function of the protein. The work would also be strengthened if a ssDNA binding mutant was made and tested given the authors interest in defining the UBL domains. For technical reasons, we did not manage to generate an inducible ssDNA binding mutant cell line to test this possibility.

August 29, 2023

Re: Life Science Alliance manuscript #LSA-2023-02111-TR

Dr. Constance Alabert
University of Dundee
MCDB
Dow Street
Dundee, Scotland DD15EH
United Kingdom

Dear Dr. Alabert,

Thank you for submitting your revised manuscript entitled "FAM111A regulates replication origin activation and cell fitness" to Life Science Alliance. The manuscript has been seen by the original reviewers whose comments are appended below. While the reviewers continue to be overall positive about the work in terms of its suitability for Life Science Alliance, some important issues remain.

Our general policy is that papers are considered through only one revision cycle; however, given that the suggested changes are relatively minor, we are open to one additional short round of revision. Please note that I will expect to make a final decision without additional reviewer input upon re-submission.

Please submit the final revision within one month, along with a letter that includes a point by point response to the remaining reviewer comments.

To upload the revised version of your manuscript, please log in to your account: <https://lsa.msubmit.net/cgi-bin/main.plex>
You will be guided to complete the submission of your revised manuscript and to fill in all necessary information.

B. MANUSCRIPT ORGANIZATION AND FORMATTING:

Sincerely,

Reviewer #1 (Comments to the Authors (Required)):

In this revised manuscript, the authors included several experiments to address some of the concerns raised by the reviewers. The interaction between FAM111A and FAM111B was verified by Co-IP experiments (however, refer to Major point #2). Additionally, a PIP mutant has been included to demonstrate that PCNA interaction is important, but not essential, for ssDNA formation in response to FAM111A overexpression. Furthermore, revisions have been made to the main text in order to enhance the accuracy of the statements. Although certain comments could not be addressed due to time and expertise limitations, the authors responded by acknowledging alternative possibilities.

The revision has improved the manuscript. If the following points listed below are addressed, it would be suitable for publication.

Major points:

1. The conclusion that "FAM111A promotes origin activation" is problematic in the absence of evidence for FAM111A's presence at origins and without proposed mechanisms. It would be more appropriate to describe the observation (reduced origin firing upon FAM111A depletion) rather than interpreting it as the function of FAM111A. This adjustment applies to various statements, including lines 84, 107, 129, 306, and 354.
2. Fig. S3C: The co-immunoprecipitation experiment utilized no antibody as a negative control. To eliminate the possibility that the band in the FAM111B Western blot results from antibody background, a negative control IP should be performed using control IgG (similar to Fig. S3B). A more effective approach to rule out this potential issue is to use FAM111B knockdown cells.

Minor points:

1. Page 6, line 167: Should Fig. S1E be Fig. S2E?
2. Page 7, line 233: Should Fig. S3E be Fig. S4E?

Reviewer #2 (Comments to the Authors (Required)):

The revised manuscript has addressed my question and therefore I am happy with it.
Some minor comments below:

In line 68 the authors state that "FAM111A promotes origin and dormant origin activation". I would suggest to revise that and state "FAM111A supports ...".

Question - could FAM111A be required for activation of dormant origins in a specific chromatin context? Thus, was the Cdc45 signal weaker in DAPI dense areas?

The authors state that Rif1 is hyperphosphorylated upon FAM111A, which could potentially impact helicase activation (e.g. Hiraga et al 2017). However, the authors also state that "none of the phosphorylation sites identified have known roles in initiation of DNA replication." - Could this statement be substantiated with a citation?

Reviewer #3 (Comments to the Authors (Required)):

I believe that the authors adequately addressed majority of my concerns either through experiments, changes/clarifications in the text. For the minority of the experiments, they have explained why particular experiments were not feasible in a revision time frame.

Reviewer #1 (Comments to the Authors (Required)):

In this revised manuscript, the authors included several experiments to address some of the concerns raised by the reviewers. The interaction between FAM111A and FAM111B was verified by Co-IP experiments (however, refer to Major point #2). Additionally, a PIP mutant has been included to demonstrate that PCNA interaction is important, but not essential, for ssDNA formation in response to FAM111A overexpression. Furthermore, revisions have been made to the main text in order to enhance the accuracy of the statements. Although certain comments could not be addressed due to time and expertise limitations, the authors responded by acknowledging alternative possibilities.

The revision has improved the manuscript. If the following points listed below are addressed, it would be suitable for publication.

Major points:

1. The conclusion that "FAM111A promotes origin activation" is problematic in the absence of evidence for FAM111A's presence at origins and without proposed mechanisms. It would be more appropriate to describe the observation (reduced origin firing upon FAM111A depletion) rather than interpreting it as the function of FAM111A. This adjustment applies to various statements, including lines 84, 107, 129, 306, and 354. Adjustments are now included in the revised manuscript line 83, 108, and 131, 306 and 354. Moreover, we have modified line 68 as suggested by Reviewer 2 and line 168 for consistency throughout the manuscript.

2. Fig. S3C: The co-immunoprecipitation experiment utilized no antibody as a negative control. To eliminate the possibility that the band in the FAM111B Western blot results from antibody background, a negative control IP should be performed using control IgG (similar to Fig. S3B). A more effective approach to rule out this potential issue is to use FAM111B knockdown cells.

In the co-immunoprecipitation experiment, Rabbit IgG was used as a control. The heavy chain of IgG (band at 50 to 55kDa) can be seen in the uncut ponceau of Fig. S3C (Fig. S6-4). The IgG band is also visible in the uncut FAM111A blot, as a secondary antibody against rabbit was used. For comparison, and as noted by Reviewer 1, the IgG band is also visible in the IgG control of the FAM111A blot of Figure S3B (Fig. S6-3).

All uncut blots were available in the revised version of the manuscript in Figure S6 and are also shown in the figure attached. We are very sorry the co-immunoprecipitation was not properly labelled. This is now corrected in the new version of the figure S3.

Minor points:

1. Page 6, line 167: Should Fig. S1E be Fig. S2E?

2. Page 7, line 233: Should Fig. S3E be Fig. S4E?

Thank you for pointing these mistakes. They have been corrected in the revised version of the manuscript.

Figure S3C

New labelling for Figure S3C

Figure S6-4 (uncut ponceau from Figure S3C)

Figure S6-4 (uncut FAM111A blot from Figure S3C)

Figure S3B

Figure S6-3 (uncut FAM111A blot from Figure S3B)

Reviewer #2 (Comments to the Authors (Required)):

The revised manuscript has addressed my question and therefore I am happy with it. Some minor comments below:

In line 68 the authors state that "FAM111A promotes origin and dormant origin activation". I would suggest to revise that and state "FAM111A supports ...".
The sentence has been modified accordingly in the revised version of the manuscript.

Question - could FAM111A be required for activation of dormant origins in a specific chromatin context? Thus, was the Cdc45 signal weaker in DAPI dense areas?
Using DAPI content by QIBC, we have quantified CDC45 level in early and late S phase, corresponding to euchromatin and heterochromatin (DAPI dense area). As we did not observe any peculiar bias upon FAM111A depletion, we decided to not include this additional analysis in the manuscript.

The authors state that Rif1 is hyperphosphorylated upon FAM111A, which could potentially impact helicase activation (e.g. Hiraga et al 2017). However, the authors also state that "none of the phosphorylation sites identified have known roles in initiation of DNA replication." - Could this statement be substantiated with a citation?

Thank you for pointing this out. We have now specified the phosphorylation sites and added the references in Line 226.

Reviewer #3 (Comments to the Authors (Required)):

I believe that the authors adequately addressed majority of my concerns either through experiments, changes/clarifications in the text. For the minority of the experiments, they have explained why particular experiments were not feasible in a revision time frame.

Thank you.

September 11, 2023

RE: Life Science Alliance Manuscript #LSA-2023-02111-TRR

Dr. Constance Alabert
University of Dundee
MCDB
Dow street
Dundee, Scotland DD15EH
United Kingdom

Dear Dr. Alabert,

Thank you for submitting your revised manuscript entitled "FAM111A regulates replication origin activation and cell fitness". We would be happy to publish your paper in Life Science Alliance pending final revisions necessary to meet our formatting guidelines.

- please add a Running Title to our system
- please add ORCID ID for the secondary corresponding--they should have received instructions on how to do so
- please add the Twitter handle of your host institute/organization as well as your own or/and one of the authors in our system
- please consult our manuscript preparation guidelines <https://www.life-science-alliance.org/manuscript-prep> and make sure your manuscript sections are in the correct order
- please add your main and supplementary figure legends to the main manuscript text after the references section
- please upload one file per figure
- please be sure to add all authors to the Authors' Contribution section
- Angus Lamond has not been added to the Author Contributions section. In addition, the contributions indicated during submission do not qualify a contributor for authorship. Please either update, or let us know if the author should be removed.

A. FINAL FILES:

B. MANUSCRIPT ORGANIZATION AND FORMATTING:

Sincerely,

September 19, 2023

RE: Life Science Alliance Manuscript #LSA-2023-02111-TRRR

Dr. Constance Alabert
University of Dundee
MCDB
Dow street
Dundee, Scotland DD15EH
United Kingdom

Dear Dr. Alabert,

Thank you for submitting your Research Article entitled "FAM111A regulates replication origin activation and cell fitness". It is a pleasure to let you know that your manuscript is now accepted for publication in Life Science Alliance. Congratulations on this interesting work.

DISTRIBUTION OF MATERIALS:

Again, congratulations on a very nice paper. I hope you found the review process to be constructive and are pleased with how the manuscript was handled editorially. We look forward to future exciting submissions from your lab.

Sincerely,
